# NON-CLASHING TEACHING IN GRAPHS: ALGORITHMS, COMPLEXITY, AND BOUNDS

**Sujoy Bhore**
Indian Institute of Technology
Bombay, India
sujoy@cse.iitb.ac.in

**Liana Khazaliya**
Technische Universität Wien
Vienna, Austria
lkhazaliya@ac.tuwien.ac.at

**Fionn Mc Inerney**
Telefónica Scientific Research
Barcelona, Spain
fmcinern@gmail.com

## ABSTRACT

Kirkpatrick et al. [ALT 2019] and Fallat et al. [JMLR 2023] introduced non-clashing teaching and proved that it is the most efficient batch machine teaching model satisfying the collusion-avoidance benchmark established in the seminal work of Goldman and Mathias [COLT 1993]. Recently, (positive) non-clashing teaching was thoroughly studied for balls in graphs, yielding numerous algorithmic and combinatorial results. In particular, Chalopin et al. [COLT 2024] and Ganian et al. [ICLR 2025] gave an almost complete picture of the complexity landscape of the positive variant, showing that it is tractable only for restricted graph classes due to the non-trivial nature of the problem and concept class.

In this work, we consider (positive) non-clashing teaching for closed neighborhoods in graphs. This concept class is not only extensively studied in various related contexts, but it also exhibits broad generality, as any finite binary concept class can be equivalently represented by a set of closed neighborhoods in a graph. In comparison to the works on balls in graphs, we provide improved algorithmic results, notably including FPT algorithms for more general classes of parameters, and we complement these results by deriving stronger lower bounds. Lastly, we obtain combinatorial upper bounds for wider classes of graphs.

## 1 INTRODUCTION

In classic machine learning models such as PAC-learning, a learner must reconstruct a concept $C$ from a concept class $\mathcal{C}$ using a random sample of examples. In contrast, in machine teaching, the examples are selected from the domain by a teacher whose goal is to aid the learner. Specifically, the teacher wants to minimize the number of labeled examples shown to the learner such that the learner can uniquely reconstruct the concept. Machine teaching is a core area in learning theory, with several models proposed in the literature (Shinohara & Miyano, 1991; Goldman & Kearns, 1995; Goldman & Mathias, 1996; Balbach, 2008; Zilles et al., 2011; Gao et al., 2016; 2017; Mansouri et al., 2019). Moreover, it has many applications in machine learning, such as sample compression (Kuzmin & Warmuth, 2007; Doliwa et al., 2014; Moran et al., 2015; Chalopin et al., 2022), inverse reinforcement learning (Ho et al., 2016; Brown & Niekum, 2019), training data security (Mei & Zhu, 2015; Zhang et al., 2018), and human-robot interaction (Thomaz & Cakmak, 2009; Akgun et al., 2012).

In this work, we focus on the *non-clashing teaching model* (Kirkpatrick et al., 2019; Fallat et al., 2023), a batch teaching framework where the teacher presents a set of examples to the learner simultaneously. This model has gained significant attention in recent years, since it was shown (Kirkpatrick et al., 2019; Fallat et al., 2023) to require the fewest examples among batch teaching models that adhere to the Goldman-Mathias collusion-avoidance criterion (Goldman & Mathias, 1996), a fundamental principle that prevents teacher-learner collusion (e.g., through encoding).

In *non-clashing teaching*, given a finite binary concept class $\mathcal{C}$, a teacher maps each concept $C \in \mathcal{C}$ to a labeled set of examples $T(C)$ that is consistent with $C$ and called the teaching set of $C$. The teaching map $T$ must satisfy the non-clashing condition: for each pair of distinct concepts $C, C' \in \mathcal{C}$, at least one of the examples in $T(C) \cup T(C')$ must be consistent with exactly one of the two concepts $C$ and $C'$. If $T$ satisfies this condition and $\max_{C \in \mathcal{C}} |T(C)| = k$, then $T$ is a *non-clashing teaching map* (NCTM) of size $k$ for $\mathcal{C}$. Notably, the teacher needs to present at most $k$ examples

(specifically, $T(C)$) for the learner to uniquely reconstruct any concept $C \in \mathcal{C}$. The non-clashing teaching dimension of $\mathcal{C}$, denoted by $\mathrm{NCTD}(\mathcal{C})$, is the smallest size of an NCTM for $\mathcal{C}$.

Machine teaching is often studied in the setting where the teacher is restricted to presenting only positively-labeled examples. Research in this setting dates back to 1980 (Angluin, 1980a;b) and has applications in, e.g., recommendation systems (Schwab et al., 2000), grammatical inference (Stolcke & Omohundro, 1994; Denis, 2001), and bioinformatics (Wang et al., 2006; Yousef et al., 2008). Notably, when $T(C)$ may only consist of positively-labeled examples for all $C \in \mathcal{C}$, the corresponding *positive non-clashing teaching dimension* of $\mathcal{C}$, denoted by $\mathrm{NCTD}^+(\mathcal{C})$, has been extensively studied (Kirkpatrick et al., 2019; Fallat et al., 2023; Chalopin et al., 2024; Ganian et al., 2025).

The initial works (Kirkpatrick et al., 2019; Fallat et al., 2023) on non-clashing teaching studied the relationship between the NCTD and the *VC-dimension* (VCD), due to its value in learning theory. A key open question of theirs is whether there is a concept class $\mathcal{C}$ such that $\mathrm{NCTD}(\mathcal{C}) > \mathrm{VCD}(\mathcal{C})$. It was also shown that deciding whether $\mathrm{NCTD}^+(\mathcal{C}) \leq 1$ ($\mathrm{NCTD}(\mathcal{C}) \leq 1$, resp.) is NP-complete when $\mathcal{C}$ is a set of open neighborhoods in a graph $G$ (Kirkpatrick et al., 2019). Although unstated, by those results, unless the Exponential Time Hypothesis (ETH)[1] fails, there is no algorithm running in $2^{o(f(k) \cdot \sqrt{|V(G)|})}$ time for either problem for any computable function $f$ of the solution size $k$.[2]

It is known that any finite binary concept class $\mathcal{C}$ can be equivalently represented by a set of balls in a graph $G$ (see Fig. 1), where the ball of radius $r \in \mathbb{N}$ centered at a vertex $v \in V(G)$ is the set $B_r(v) := \{u \mid \mathrm{dist}(u, v) \leq r, \ u \in V(G)\}$ (Chalopin et al., 2023). This was one of two main motivations for the recent works (Chalopin et al., 2024; Ganian et al., 2025) on *non-clashing teaching* for balls in graphs. The other was that this is a natural and fundamental concept class that has been investigated for related topics like sample compression (Chalopin et al., 2023), and its VC-dimension is well-studied (Chepoi et al., 2007; Bousquet & Thomassé, 2015; Ducoffe et al., 2020).

Given a graph $G$, a set of balls $\mathcal{B}$ in $G$, and $k \in \mathbb{N}$, NON-CLASH asks whether $\mathrm{NCTD}^+(\mathcal{B}) \leq k$. If $\mathcal{B}$ contains every ball in $G$, then this problem is known as STRICT NON-CLASH. When $G$ is restricted to the class of (co-)bipartite or split graphs, STRICT NON-CLASH remains NP-hard (Chalopin et al., 2024), even if $k = 2$ for the latter (Ganian et al., 2025). STRICT NON-CLASH can be solved in $2^{2^{\mathcal{O}(\mathrm{vc}(G))}} \cdot |V(G)|^{\mathcal{O}(1)}$ time, where $\mathrm{vc}(G)$ is the vertex cover number of $G$, and this is tight under the ETH (Chalopin et al., 2024). This FPT[3] algorithm was improved upon by considering a more general parameter and problem, as it was shown that NON-CLASH is FPT parameterized by the vertex integrity of $G$ (Ganian et al., 2025). As NON-CLASH is W[1]-hard[4] parameterized by the feedback vertex number, pathwidth, and solution size $k$ combined (Ganian et al., 2025), these results give a nearly complete picture of its complexity landscape. There is a $2^{\mathcal{O}(|V(G)| \cdot d \cdot k \cdot \log |V(G)|)}$-time algorithm for NON-CLASH, and unless the ETH fails, there is no $2^{o(|V(G)| \cdot d \cdot k)}$-time algorithm, even for STRICT NON-CLASH, where $d$ is the diameter of $G$ (Ganian et al., 2025). Finally, (near-)optimal NCTMs are known for balls if $G$ is a tree, cycle, cactus or interval graph (Chalopin et al., 2024).

**Our Contributions.** We explore non-clashing teaching for balls in graphs from a combinatorial and computational perspective, providing a comprehensive understanding of the problem. As discussed above, there is an inherent algorithmic intractability (NP-hard and W[1]-hard). With the goal of obtaining improved algorithmic results for larger graph classes under more general parameters, we study both variants of non-clashing teaching for closed neighborhoods (balls of radius 1) in graphs:

---

N-NCTD
**Input:** A graph $G$, a set $\mathcal{B}$ of closed neighborhoods in $G$, and a positive integer $k$.
**Question:** Is $\mathrm{NCTD}(\mathcal{B}) \leq k$?

---

N-NCTD$^+$
**Input:** A graph $G$, a set $\mathcal{B}$ of closed neighborhoods in $G$, and a positive integer $k$.
**Question:** Is $\mathrm{NCTD}^+(\mathcal{B}) \leq k$?

---

[1] Roughly, the ETH states that $n$-variable and $m$-clause 3-SAT cannot be solved in $2^{o(n+m)}$ time.

[2] Their results are proven by reductions from a matching problem in graphs, and they prove that problem is NP-hard by a reduction from $n$-variable and $m$-clause 3-SAT that produces a graph with $\Theta(n \cdot m)$ vertices.

[3] A problem with input size $n$ is fixed-parameter tractable (FPT) parameterized by parameter $p \in \mathbb{N}$ if it admits an $f(p) \cdot n^{\mathcal{O}(1)}$-time algorithm (i.e., an FPT algorithm) for some computable function $f$.

[4] A problem that is W[1]-hard parameterized by $p$ is not FPT parameterized by $p$ unless W[1] = FPT.

We emphasize that this is as general as studying finite binary concept classes. In fact, the afore-mentioned representation of any finite binary concept class $\mathcal{C} \subseteq 2^V$ by a set of balls in a graph $G$, holds for a set of closed neighborhoods in $G$. Indeed, $V(G) = V \cup \{x_C \mid C \in \mathcal{C}\}$, the $x_C$ vertices form a clique, $x_C$ is adjacent to $v \in V$ if and only if $v \in C$, and $\mathcal{B} = \{N[x_C] \mid C \in \mathcal{C}\}$. See Fig. 1. It is also motivated by the plethora of works that have considered the *VC-dimension* of *closed neighborhoods* in graphs (Haussler & Welzl, 1987; Anthony et al., 1995; Kranakis et al., 1997; Bousquet et al., 2015; Ducoffe et al., 2020; Ducoffe, 2021; Chalopin et al., 2023; Coudert et al., 2024). See (Foucaud et al., 2025) for a list of references on other graph-based concept classes.

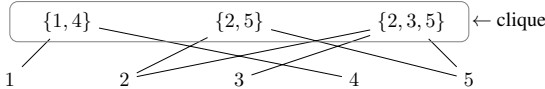

Figure 1: Graph $G$ for a binary concept class $\mathcal{C} = \{\{1,4\}, \{2,5\}, \{2,3,5\}\}$ and $\mathcal{B} = \{N[C]\}_{C \in \mathcal{C}}$.

In Section 2, we present improved algorithmic upper and lower bounds, which match in the case of N-NCTD$^+$, unlike in the case of NON-CLASH. Specifically, no algorithmic lower bound was previously known for the NCTD of balls in graphs, however, we prove that, unless the ETH fails, there is no $2^{o(f(k) \cdot |V(G)|)}$-time algorithm for any computable function $f$ (Theorem 2). This also dramatically improves upon the only previously known lower bound for the NCTD of a concept class, which is that, unless the ETH fails, no $2^{o(f(k) \cdot \sqrt{|V(G)|})}$-time algorithm exists (Kirkpatrick et al., 2019). Further, our lower bound is almost tight due to the $2^{\mathcal{O}(k \cdot |V(G)| \cdot \log |V(G)|)}$-time algorithm that can be inferred from (Ganian et al., 2025). We then show that N-NCTD$^+$ can be solved in $2^{\mathcal{O}(|E(G)|)}$ time (Theorem 4), and unless the ETH fails, this is tight, i.e., there is no $2^{o(|E(G)|)}$-time algorithm (Theorem 3); in fact there is even no $2^{o(f(k) \cdot (|V(G)| + |E(G)|))}$-time algorithm for any computable function $f$. This is also a new strong lower bound for NON-CLASH.

In Section 3, we provide two highly non-trivial algorithms to prove that N-NCTD$^+$ is FPT parameterized by the *treedepth* of $G$ (Theorem 5) and N-NCTD is FPT parameterized by the *vertex cover number* of $G$ (Theorem 7). These are two well-established structural graph parameters that have been considered in, e.g., Bayesian network learning (Korhonen & Parviainen, 2015; Harviainen & Koivisto, 2023) and other areas of AI (Ganian et al., 2020; Eiben et al., 2023). Notably, we again achieve positive results that were not obtained in the setting of balls in graphs as *treedepth* is a more general parameter than *vertex integrity*,[5] and no FPT algorithm parameterized by a structural graph parameter was known for any concept class when the teacher may use negatively-labeled examples.

Finally, in Section 4, we obtain combinatorial bounds for larger graph classes than those in (Chalopin et al., 2024). Exploiting forbidden minors, we prove that, if $\mathcal{B}$ is a set of closed neighborhoods in a planar graph $G$, then NCTD$^+(\mathcal{B}) \leq 7$ (Theorem 12) and NCTD$(\mathcal{B}) \leq 5$ (Theorem 13). Employing geometric arguments, we prove that, if $\mathcal{B}$ is a set of closed neighborhoods in a unit square graph $G$, then NCTD$^+(\mathcal{B}) \leq 4$ (Theorem 14). See Table 1 for a summary of our results and the literature.

Table 1: Summary of our results and the literature. NP-h, CN, $V$, $E$, [1], and [2] denote NP-hard, closed neighborhoods, $V(G)$, $E(G)$, (Ganian et al., 2025), and (Chalopin et al., 2024), respectively.

| | NCTD$^+$ Balls | NCTD$^+$ CN | NCTD Balls | NCTD CN |
|---|---|---|---|---|
| **Solution size k** | NP-h if $k=1$ (Th. 3) | NP-h if $k=1$ (Th. 3) | NP-h if $k=1$ (Th. 2) | NP-h if $k=1$ (Th. 2) |
| **Vertex cover** | FPT [1] | FPT (Th. 5) | ? | FPT (Th. 7) |
| **Vertex integrity** | FPT [1] | FPT (Th. 5) | ? | ? |
| **Treedepth** | ? | FPT (Th. 5) | ? | ? |
| **Treewidth** | W[1]-hard [1] | ? | ? | ? |
| **Alg. up. bound** | $2^{\mathcal{O}(k \cdot d \cdot |V| \cdot \log |V|)}$ [1] | $2^{\mathcal{O}(|E|)}$ (Th. 4) | $2^{\mathcal{O}(k \cdot d \cdot |V| \cdot \log |V|)}$ [1] | $2^{\mathcal{O}(k \cdot |V| \cdot \log |V|)}$ [1] |
| **Alg. low. bound** | $2^{o(k \cdot d \cdot |V|)}$ [1] | $2^{o(f(k) \cdot (|V| + |E|))}$ (Th. 3) | $2^{o(f(k) \cdot |V|)}$ (Th. 2) | $2^{o(f(k) \cdot |V|)}$ (Th. 2) |
| **Planar** | Unbounded by cycle [2] | $\leq 7$ (Th. 12) | $\leq 615$ [2] | $\leq 5$ (Th. 13) |
| **Unit square** | Unbounded by cycle [2] | $\leq 4$ (Th. 14) | $\leq 615$ [2] | $\leq 4$ (Th. 14) |

---

[5]For any problem on a graph $G$, if it admits an FPT algorithm parameterized by the treedepth of $G$, then it admits an FPT algorithm parameterized by the vertex integrity of $G$, but not the other way round.

## 2 IMPROVED ALGORITHMIC UPPER AND LOWER BOUNDS

In this section, we first prove an algorithmic lower bound for N-NCTD that is stronger than the previous one from the literature, and then tight algorithmic bounds for N-NCTD$^+$. We begin by introducing problem-specific terminology. Let $G$ be a graph, $\mathcal{B}$ a set of closed neighborhoods in $G$, and $T$ a teaching map for $\mathcal{B}$. If, for all $B \in \mathcal{B}$, $T(B) \subseteq B$, then $T$ is a *positive teaching map* for $\mathcal{B}$. In either case, $T$ is *non-clashing* for $\mathcal{B}$ if, for all distinct $N[u], N[v] \in \mathcal{B}$, there exists $w \in T(N[u]) \cup T(N[v])$ such that $w \in N[u] \cup N[v]$ and $w \notin N[u] \cap N[v]$; we say that $w$ *distinguishes* $N[u]$ and $N[v]$. For all $n \in \mathbb{Z}^+$, we set $[n] := \{1, \ldots, n\}$.

From the exact exponential algorithm for NON-CLASH in (Ganian et al., 2025), it can be observed that N-NCTD can be solved in $2^{\mathcal{O}(k \cdot |V(G)| \cdot \log |V(G)|)}$ time. In the next theorem, we provide an improved lower bound, via a reduction from 3-SAT, that almost matches this upper bound. However, we first need the following lemma, which will be useful for the proof of that theorem. Note that, given a graph $G$, two distinct vertices $u, v \in V(G)$ are *false twins* if and only if $N(u) = N(v)$.

**Lemma 1.** *Let $G$ be a graph with $4$ pairwise false twins $u_1, \ldots, u_4$ with $N[u_1], \ldots, N[u_4] \in \mathcal{B}$. Then, for any NCTM $T$ of size $1$ for $\mathcal{B}$, there exists $i \in [4]$ such that $T(N[u_i]) = \{u_i\}$.*

**Theorem 2.** *Unless the ETH fails,* N-NCTD *cannot be solved in $2^{o(f(k) \cdot |V(G)|)}$ time for any computable function $f$.*

*Proof.* Given an instance $\varphi$ of $n$-variable and $m$-clause 3-SAT, we construct a graph $G$ as follows:

1. for each variable $x_i$ and clause $C_j$ in $\phi$, there is a clause vertex $c_j$, a variable vertex $v_i$, and two literal vertices $t_i$ and $f_i$ that are both adjacent to $v_i$;

2. there is an additional dummy variable vertex $v_0$ that has no adjacent literal vertices;

3. for all $i \in \{0\} \cup [n]$, there is a set of vertices $\mathcal{V}_i = \{v_i^0, v_i^1, v_i^2, v_i^3, v_i^4\}$ and a special vertex $v_i^\star$, where $v_i^\star$ is adjacent to each vertex in $\mathcal{V}_i$, and each vertex in $\mathcal{V}_i \cup \{v_i^\star\}$ is adjacent to $v_i$;

4. for all $i \in [n]$, if the literal $x_i$ ($\overline{x}_i$, resp.) is in the clause $C_j$ in $\phi$, then the literal vertex $f_i$ ($t_i$, resp.) is adjacent to the clause vertex $c_j$ and there is a vertex $c_{j,i}$ adjacent to $v_i$, $c_j$, $v_i^\star$, and each vertex in $\mathcal{V}_i$;

5. the set $\{c_j\}_{j \in [m]} \cup \{v_i\}_{i \in \{0\} \cup [n]}$ forms a clique.

This concludes the construction of $G$. We set $\mathcal{B}$ to contain the closed neighborhood of each vertex in $\bigcup_{j \in [m]} \{c_j\}$ and $\bigcup_{i \in \{0\} \cup [n]} (\{v_i, v_i^\star\} \cup \{v_i^p\}_{p \in [4]})$ (note that, for all $i \in \{0\} \cup [n]$, $N[v_i^0] \notin \mathcal{B}$). Thus, from $\varphi$, the reduction outputs the N-NCTD instance $(G, \mathcal{B}, 1)$.

To show the correctness of the reduction in the first direction, let $T$ be an NCTM of size $1$ for $\mathcal{B}$. By Lemma 1, for all $i \in \{0\} \cup [n]$, there exists $p \in [4]$ such that $T(N[v_i^p]) = \{v_i^p\}$, say, w.l.o.g., $T(N[v_i^1]) = \{v_i^1\}$. Hence, for all $i \in \{0\} \cup [n]$, $T(N[v_i^\star]) \subset \mathcal{V}_i$ since $N[v_i^1] \subset N[v_i^\star]$. Therefore, for all $i \in \{0\} \cup [n]$, $T(N[v_i]) \subset (N[v_i] \setminus N[v_i^\star])$ since $N[v_i^\star] \subset N[v_i]$. As $T(N[v_0])$ is not adjacent to any literal vertices, then it must consist of a clause or variable vertex. As the clause and variable vertices form a clique, for all $i \in [n]$, $T(N[v_i])$ cannot consist of a clause or variable vertex, and hence, $T(N[v_i]) \subset \{t_i, f_i\}$. We extract a variable assignment $\tau \colon \{x_1, \ldots, x_n\} \to \{\texttt{True}, \texttt{False}\}$ as follows: if $T(N[v_i]) = \{t_i\}$, then $\tau(x_i) := \texttt{True}$, and otherwise, $\tau(x_i) := \texttt{False}$.

Toward a contradiction, suppose that $\tau$ is not a satisfying assignment for $\varphi$. Then, there exists a clause $C_j$ in $\varphi$ that is not satisfied by $\tau$. Let $x$, $x'$, and $x''$ be the 3 variables in $C_j$ in $\varphi$ (and the corresponding variable vertices in $G$). By the construction of $G$ and the definition of $\tau$, as $C_j$ is not satisfied, $T(N[x]), T(N[x']), T(N[x'']) \subset N[c_j]$. Thus, as $T$ is non-clashing for $N[c_j]$ and each of $N[x], N[x']$, and $N[x'']$, then $T(N[c_j])$ consists of a vertex $u \in V(G)$, where either *(1)* $u \in N[c_j]$, $u \notin N[x]$, $u \notin N[x']$, and $u \notin N[x'']$, or *(2)* $u \notin N[c_j]$, $u \in N[x]$, $u \in N[x']$, and $u \in N[x'']$. No such vertex $u$ exists, contradicting that $T$ is an NCTM for $\mathcal{B}$. So, $\tau$ is a satisfying assignment for $\varphi$.

For the reverse direction, let $\tau \colon \{x_1, \ldots, x_n\} \to \{\texttt{True}, \texttt{False}\}$ be a satisfying assignment of the given 3-SAT formula $\varphi$. We define a teaching map $T$ for $\mathcal{B}$ as follows. For all $i \in [n]$, if $\tau(x_i) = \texttt{True}$, then $T(N[v_i]) := \{t_i\}$, and otherwise, $T(N[v_i]) := \{f_i\}$. For all $i \in \{0\} \cup [n]$ and $p \in [4]$, $T(N[v_i^p]) := \{v_i^p\}$. For all $i \in \{0\} \cup [n]$, $T(N[v_i^\star]) := \{v_i^0\}$. For all $j \in [m]$, there exists $i \in [n]$ such that $\tau(x_i)$ satisfies $C_j$, and we set $T(N[c_j]) := \{c_{j,i}\}$. Finally, $T(N[v_0]) := \{v_1\}$.

We prove that $T$ is an NCTM for $\mathcal{B}$. For all $i \in \{0\} \cup [n]$ and $p \in [4]$, $T(N[v_i^p]) = \{v_i^p\}$ distinguishes $N[v_i]$ and $N[w]$ for all $w \in V(G)$ such that $N[w] \in \mathcal{B} \setminus \{N[v_i], N[v_i^\star]\}$. For all $i \in \{0\} \cup [n]$ and $p \in [4]$, $T(N[v_i])$ distinguishes $N[v_i]$ and $N[v_i^p]$, as well as $N[v_i]$ and $N[v_i^\star]$, as $T(N[v_i]) \subset N[v_i]$, $T(N[v_i]) \cap N[v_i^p] = \emptyset$, and $T(N[v_i]) \cap N[v_i^\star] = \emptyset$. For all $i \in \{0\} \cup [n]$ and $p \in [4]$, $T(N[v_i^\star])$ distinguishes $N[v_i^\star]$ and $N[v_i^p]$ since $T(N[v_i^\star]) \subset N[v_i^\star]$ and $T(N[v_i^\star]) \cap N[v_i^p] = \emptyset$. Thus, all pairs of closed neighborhoods containing $N[v_i^p]$ for some $i \in \{0\} \cup [n]$ and $p \in [4]$ are distinguished. For all $i \in \{0\} \cup [n]$, $N[v_i^\star]$ and any other closed neighborhood in $\mathcal{B}$ are distinguished since $T(N[v_i^\star]) \subset N[v_i^\star]$, but $T(N[v_i^\star])$ is not a subset of any closed neighborhood in $\mathcal{B}$ not already covered above.

For all distinct $i \in [n]$ and $q \in \{0\} \cup [n]$, $T(N[v_i]) \subset N[v_i]$ and $T(N[v_i]) \cap N[v_q] = \emptyset$, and so, $N[v_i]$ and $N[v_q]$ are distinguished. For all distinct $j, \ell \in [m]$, $T(N[c_j]) \subset N[c_j]$ and $T(N[c_j]) \cap N[c_\ell] = \emptyset$, and so, $N[c_j]$ and $N[c_\ell]$ are distinguished. For all $i \in [n]$ and $j \in [m]$ such that $c_{j,i} \in T(N[c_j])$, $\tau(x_i)$ satisfies the clause $C_j$ in $\varphi$, and so, $T(N[v_i]) \cap N[c_j] = \emptyset$, resulting in $N[v_i]$ and $N[c_j]$ being distinguished by $T(N[v_i])$. For all $i \in [n]$ and $j \in [m]$ such that $c_{j,i} \notin T(N[c_j])$, $N[v_i]$ and $N[c_j]$ are distinguished by $T(N[c_j])$. Lastly, for all $j \in [m]$, $T(N[c_j])$ distinguishes $N[c_j]$ and $N[v_0]$. This completes the case analysis, and thus, $T$ is an NCTM for $\mathcal{B}$.

Lastly, if N-NCTD can be solved in $2^{o(f(k) \cdot |V(G)|)}$ time for a computable function $f$, then 3-SAT can be solved in $2^{o(n+m)}$ time as $f(k) = \mathcal{O}(1)$ and $|V(G)| = \mathcal{O}(n+m)$, contradicting the ETH. $\square$

A similar reduction proves the following lower bound, which we then prove is tight under the ETH.

**Theorem 3.** *Unless the* ETH *fails,* N-NCTD$^+$ *cannot be solved in* $2^{o(f(k) \cdot (|V(G)|+|E(G)|))}$ *time for any computable function* $f$.

**Theorem 4.** N-NCTD$^+$ *can be solved in* $2^{\mathcal{O}(|E(G)|)}$ *time.*

*Proof.* For all $v \in V(G)$, $|N[v]| = d(v) + 1$, where $d(v)$ is the degree of $v$. Therefore, there are $2^{d(v)+1}$ choices for $T(N[v])$. So, there are at most $2^{\sum_{v \in V(G)} (d(v)+1)} = 2^{\mathcal{O}(|E(G)|)}$ distinct positive teaching maps for $\mathcal{B}$, each of which can be checked to be non-clashing or not in polynomial time. $\square$

## 3 FIXED-PARAMETER ALGORITHMS

In this section, we prove that N-NCTD$^+$ is FPT parameterized by the *treedepth* of $G$ and N-NCTD is FPT parameterized by the vertex cover number of $G$. As mentioned before, this further motivates considering closed neighborhoods instead of balls in graphs, as treedepth is a more general parameter than vertex integrity (see footnote 5), and no FPT algorithm was known when the teacher may use negatively-labeled examples. The *treedepth* $\mathtt{td}(G)$ of a connected graph $G$ is the minimum height of a rooted tree $\mathcal{T}$ such that $V(\mathcal{T}) = V(G)$ and, for all $uv \in E(G)$, either $u$ is an ancestor of $v$ or $v$ is an ancestor of $u$ in $\mathcal{T}$. The tree $\mathcal{T}$ is a *treedepth decomposition* of $G$ (see Fig. 2 for an illustration).

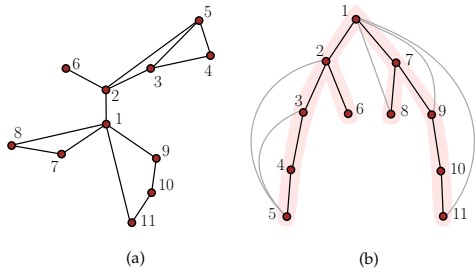

(a)            (b)

Figure 2: (a) A graph $G$. (b) A treedepth decomposition $\mathcal{T}$ of $G$ witnessing $\mathtt{td}(G) \leq 4$.

**Theorem 5.** N-NCTD$^+$ *is* FPT *parameterized by the treedepth of* $G$.

We prove Theorem 5 via an algorithm that exhaustively applies a reduction rule to prune the tree $\mathcal{T}$ in the treedepth decomposition of $G$ from the bottom up, yielding a tree whose number of vertices is a function of $\mathtt{td}(G)$ (see Fig. 4). From there, we compute NCTD$^+(\mathcal{B})$ by brute force in FPT time w.r.t. $\mathtt{td}(G)$. For simplicity, we state the reduction rule in more general terms and do not specify the component to delete. However, the proof that it is safe (Lemma 6) specifies the component; a reduction rule is *safe* if the input instance is a YES-instance if and only if the output instance is.

**Reduction Rule 1.** *Let $G$ be a graph, $X \subseteq V(G)$ a subset of its vertices, and $A = \{A_1, \ldots, A_\ell\}$ a subset of the connected components of $G - X$ such that $\max_{i \in [\ell]} |A_i| = t$. If $\ell > (|X| + t) \cdot 2^{(|X|+t)^2} \cdot 2^{2t+|X|+1}$, then delete a particular connected component in $A$.*

**Lemma 6.** *Reduction rule 1 is safe for* $\text{N-NCTD}^+$.

*Proof Sketch.* First, we give intuition. If $\ell$ is large, by the pigeonhole principle, there are 3 automorphic components in $A$ that are "identical" with respect to their adjacencies in $X$, and the closed neighborhoods of $\mathcal{B}$ that they contain along with their teaching sets. We select one of these 3 components, consider each teaching set that contains a vertex from it, and replace that vertex by one of its two automorphic copies (made possible by their identicalness), allowing to delete that component.

Formally, let $X \subseteq V(G)$ be such that $A = \{A_1, \ldots, A_\ell\}$ is a subset of the connected components of $G - X$, where $\max_{i \in [\ell]} |A_i| = t$ and $\ell > (|X|+t) \cdot 2^{(|X|+t)^2} \cdot 2^{2t+|X|+1}$. Let $T$ be a positive NCTM of size at most $k$ for $\mathcal{B}$. We first prove that there exist $A_P, A_Q, A_R \in A$ that are automorphic to each other and such that, for any distinct $A_C, A_D \in \{A_P, A_Q, A_R\}$, there exists an automorphism $\sigma : V(A_C) \to V(A_D)$ where, for all $c \in V(A_C)$ and $d \in V(A_D)$ with $\sigma(c) = d$, it holds that:

**1**. $N(c) \cap X = N(d) \cap X$;

**2**. $N[c] \in \mathcal{B}$ if and only if $N[d] \in \mathcal{B}$;

**3**. if $N[c], N[d] \in \mathcal{B}$, then $T(N[c]) \cap X = T(N[d]) \cap X$ and, for all $u \in V(A_C)$ and $v \in V(A_D)$ with $\sigma(u) = v$, we have $u \in T(N[c]) \cap V(A_C)$ if and only if $v \in T(N[d]) \cap V(A_D)$.

We prove this via the pigeonhole principle. Trivially, the number of non-automorphic graphs on at most $|X| + t$ vertices is at most $(|X| + t) \cdot 2^{(|X|+t)^2}$. Thus, since $\ell > 2(|X| + t) \cdot 2^{(|X|+t)^2}$, there exist $A_P, A_Q, A_R \in A$ that are automorphic to each other and such that, for any distinct $A_C, A_D \in \{A_P, A_Q, A_R\}$, there exists an automorphism $\sigma : V(A_C) \to V(A_D)$ where, for all $c \in V(A_C)$ and $d \in V(A_D)$ with $\sigma(c) = d$, 1. holds. Since $\ell > 2(|X| + t) \cdot 2^{(|X|+t)^2} \cdot 2^t$, we also have that 2. holds. Moreover, as $\ell > 2(|X| + t) \cdot 2^{(|X|+t)^2} \cdot 2^t \cdot 2^{t+|X|}$, we also have that 3. holds.

Select $w \in V(G) \setminus V(A_P)$ such that there exists $v \in T(N[w]) \cap V(A_P)$; we can assume $w$ exists, as otherwise we delete $V(A_P)$. Note that $w \in X$. We show that removing $v$ from $T(N[w])$ and adding a particular vertex $z \in V(A_Q) \cup V(A_R)$ to $T(N[w])$ maintains that $T$ is a positive NCTM for $\mathcal{B}$ in $G$. Let $\sigma_{P,Q} : V(A_P) \to V(A_Q)$ ($\sigma_{P,R} : V(A_P) \to V(A_R)$, resp.) be the automorphism between $V(A_P)$ and $V(A_Q)$ ($V(A_R)$, resp.). If $T(N[w]) \cap V(A_Q) = \emptyset$, then $z = \sigma_{P,Q}(v)$, and if not, but $\sigma_{P,R}(v) \notin T(N[w])$, then $z = \sigma_{P,R}(v)$, and otherwise no vertex is added to $T(N[w])$.

Let $T'$ be the teaching map for $\mathcal{B}$ obtained from $T$ by applying the above procedure for all $w \in X$ and $v \in V(A_P)$ such that $v \in T(N[w])$. Note that $T'$ has size at most $k$, and $T'$ is a positive teaching map for $\mathcal{B}$ since $T$ is, and due to property 1. above. We show that $T'$ is a positive NCTM for $\mathcal{B}$. Toward a contradiction, suppose there exist $x, y \in V(G)$ such that $T'$ does not satisfy the non-clashing condition for $N[x]$ and $N[y]$. Due to how $T'$ was obtained from $T$, and since $T$ is an NCTM for $\mathcal{B}$, while $T'$ is not, then the only vertices in $T(N[x]) \cup T(N[y])$ distinguishing $N[x]$ and $N[y]$ are in $V(A_P)$. Thus, w.l.o.g., $v \in (T(N[x]) \cup T(N[y])) \cap V(A_P)$ distinguished $N[x]$ and $N[y]$. Then, $\sigma_{P,Q}(v) \in T'(N[x]) \cup T'(N[y])$ and/or $\sigma_{P,R}(v) \in T'(N[x]) \cup T'(N[y])$, and, w.l.o.g., $v \in N[x]$ and $v \notin N[y]$. We do a case analysis (see Fig. 3). Let $u = \sigma_{P,Q}(v)$ and $u' = \sigma_{P,R}(v)$.

**Case 1:** $x \in N[v] \cap V(A_P)$. Thus, $v \in T'(N[x])$ distinguishes $N[x]$ and $N[y]$, a contradiction.

**Case 2:** $x \in N(v) \cap X$ and $y \notin N[u]$. As $N(v) \cap X = N(u) \cap X = N(u') \cap X$, then $u, u' \in N[x]$. If $u \in T'(N[x])$, then $u$ distinguishes $N[x]$ and $N[y]$, a contradiction. Otherwise, $u' \in T'(N[x])$ and there exists a vertex $c \in T'(N[x]) \cap V(A_Q)$, and so, since $y \notin N[v]$, then $y \notin N[u'] \cap X$, and thus, $N[x]$ and $N[y]$ are distinguished by $u'$ and/or $c$, a contradiction.

**Case 3:** $x \in N(v) \cap X$ and $y \in N[u]$. Thus, $y \in V(A_Q)$. Let $\sigma_{P,Q}(y') = y$. As $T$ is an NCTM for $\mathcal{B}$, there exists $r \in T(N[x]) \cup T(N[y'])$ that distinguishes $N[x]$ and $N[y']$. We now study subcases.

**Case 3.1:** $r \in T(N[y'])$. If $r \notin V(A_P)$, then $r \in T'(N[y])$ distinguishes $N[x]$ and $N[y]$, a contradiction. If $r \in V(A_P)$, then $\sigma_{P,Q}(r) \in T'(N[y])$ distinguishes $N[x]$ and $N[y]$, a contradiction.

**Case 3.2:** $r \in T(N[x])$. If $r \notin V(A_P)$, then $r \in T'(N[x])$. If $r \in V(A_P)$, then $\sigma_{P,Q}(r) \in T'(N[x])$ and/or $\sigma_{P,R}(r) \in T'(N[x])$. We again study subcases.

**Case 3.2.1:** $r \notin N[y]$. If $r \notin V(A_P)$, then $r \in T'(N[x])$ distinguishes $N[x]$ and $N[y]$, a contradiction. Otherwise, as $r \in T(N[x])$ distinguishes $N[x]$ and $N[y']$, then $r \notin N[y']$, which implies that $\sigma_{P,Q}(r) \notin N[y]$. So, $\sigma_{P,Q}(r)$ or $\sigma_{P,R}(r)$ in $T'(N[x])$ distinguishes $N[x]$ and $N[y]$, a contradiction.

**Case 3.2.2:** $r \in N[y]$. As $T$ is an NCTM for $\mathcal{B}$, there exists $r' \in T(N[x]) \cup T(N[y])$ distinguishing $N[x]$ and $N[y]$. If $r' \in T(N[y])$, then $r' \in T'(N[y])$ distinguishes $N[x]$ and $N[y]$. Hence, $r' \in T(N[x])$. Further, if $r' \notin N[y']$, then either $r' \notin V(A_P)$, and so, $r' \in T'(N[x])$, or $r' \in V(A_P)$, in which case $\sigma_{P,R}(r') \in T'(N[x])$ as $r \in T'(N[x]) \cap N[y]$. In both cases, $T'$ satisfies the non-clashing condition for $N[x]$ and $N[y]$, a contradiction. Thus, $r' \in N[y']$, and as $r \in T'(N[x]) \cap N[y]$, then $\sigma_{P,R}(r') \in T'(N[x])$, which distinguishes $N[x]$ and $N[y]$, a contradiction.

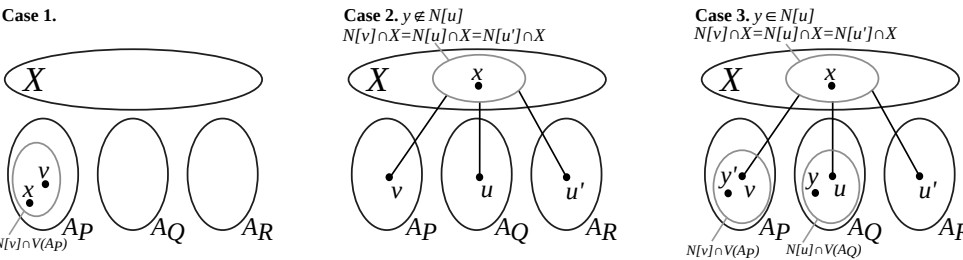

Figure 3: Cases 1, 2, and 3 in the forward direction of the proof of Lemma 6. Recall that $y \notin N[v]$.

That completes the case analysis, and thus, $T'$ is a positive NCTM for $\mathcal{B}$. Let $G' := G - A_P$ and let $\mathcal{B}'$ equal $\mathcal{B}$ restricted to $G'$. Since the only teaching sets of $T'$ containing vertices in $V(A_P)$ are those of the closed neighborhoods centered in vertices of $V(A_P)$, then $T'$ restricted to $\mathcal{B}'$ is a positive NCTM of size at most $k$ for $\mathcal{B}'$ in $G'$. This completes the first direction of the proof.

For the reverse direction, we do the (simpler) inverse of the above. Let $T'$ be a positive NCTM of size at most $k$ for $\mathcal{B}'$ in $G'$. Since there exists an automorphism $\sigma_{Q,R} : V(A_Q) \to V(A_R)$, then, for each $q \in V(A_q)$ such that $N[q] \in \mathcal{B}'$, $T'(N[q]) \cap V(A_Q) \neq \emptyset$ and $T'(N[\sigma_{Q,R}(q)]) \cap V(A_R) \neq \emptyset$. Indeed, otherwise $T'$ does not satisfy the non-clashing condition for that pair of closed neighborhoods. We extend $T'$ to a positive NCTM $T$ of size at most $k$ for $\mathcal{B}$ in $G$ as follows. For all $p \in V(A_P)$, let $p' := \sigma_{P,Q}(p)$. For all $x \in V(G')$, set $T(N[x]) := T'(N[x])$, and, for all $p \in V(A_P)$ such that $N[p] \in \mathcal{B}$, set $T(N[p]) := (T(N[p']) \cap X) \bigcup_{q \in T(N[p']) \cap V(A_Q)} \{\sigma_{P,Q}^{-1}(q)\}$. It follows that $T$ is a positive NCTM of size at most $k$ for $\mathcal{B}$ in $G$ due to the properties of $A_P$, $A_Q$, and $A_R$. $\qquad\square$

*Proof of Theorem 5.* Use the algorithm of (Reidl et al., 2014) to compute a treedepth decomposition $\mathcal{T}$ of $G$ of depth $\mathtt{td}(G)$ in $2^{O(\mathtt{td}(G)^2)} \cdot |V(G)|$ time. From the bottom of $\mathcal{T}$ up, exhaustively apply Reduction rule 1, which is safe by Lemma 6. Specifically, first consider the children of a vertex at depth $\mathtt{td}(G) - 1$. These children (leaves of $\mathcal{T}$) form $A$, and their ancestors in $\mathcal{T}$ form $X$. Exhaustively apply Reduction Rule 1 for the children of each vertex at depth $\mathtt{td}(G) - 1$. Once Reduction Rule 1 cannot be applied, each vertex at depth $\mathtt{td}(G) - 1$ has at most $g_1(\mathtt{td}(G))$ children, for a computable function $g_1$. Proceeding by induction on the depth $\mathtt{td}(G) - j$ of the considered vertices, suppose that, for some $1 \leq i < \mathtt{td}(G)$, it holds that, for all $1 \leq j \leq i$, after exhaustively applying Reduction Rule 1, each vertex at depth $\mathtt{td}(G) - j$ has at most $g_j(\mathtt{td}(G))$ children for a computable function $g_j$. Then, exhaustively applying Reduction Rule 1 at depth $\mathtt{td}(G) - i - 1$ results in each vertex at depth $\mathtt{td}(G) - i - 1$ having at most $g_{i+1}(\mathtt{td}(G))$ children for a computable function $g_{i+1}$. See Fig. 4 for an illustration of this pruning process. In total, exhaustively applying Reduction Rule 1 as above takes FPT time with respect to $\mathtt{td}(G)$ as we need to find the components $A_P$, $A_Q$, and $A_R$ each time. Once Reduction rule 1 cannot be applied at the root of $\mathcal{T}$, we have proven by induction that the remaining graph $G'$ has at most $f(\mathtt{td}(G))$ vertices, and thus, at most $f'(\mathtt{td}(G))$ edges, for computable functions $f$ and $f'$. Now, apply the algorithm of Theorem 4 on $G'$, which runs in $f''(\mathtt{td}(G))$ time for a computable function $f''$ since $|E(G')| = f'(\mathtt{td}(G))$. $\qquad\square$

We proceed with the vertex cover number parameterization for N-NCTD. The *vertex cover number* of a graph $G$ is the minimum size of a subset of vertices $X \subseteq V(G)$ such that each edge in $G$ is incident to at least one vertex in $X$. We emphasize that allowing negatively-labeled examples in the teaching sets poses a significant challenge as the problem is no longer local, which can be seen by comparing the proof of the next result with the one for STRICT NON-CLASH (Chalopin et al., 2024).

**Theorem 7.** N-NCTD *is* FPT *parameterized by the vertex cover number of $G$.*

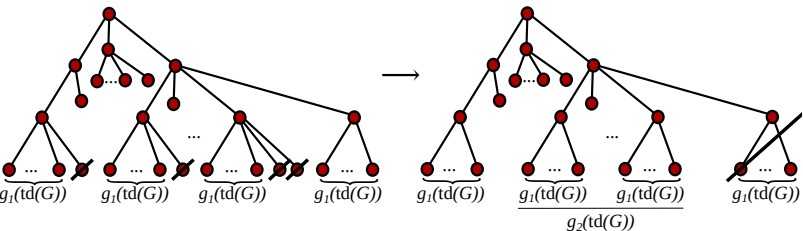

Figure 4: Pruning of a treedepth decomposition as in the proof of Theorem 5.

We prove Theorem 7 via a kernelization algorithm[6] which exhaustively applies two reduction rules. However, we first show that the solution size $k$ can be bounded above by a function of the vertex cover number of $G$ as this is a necessary condition for one of the reduction rules. To do so, we consider equivalence classes for the vertices in the independent set with respect to the vertex cover. Given a graph $G$, let $X \subseteq V(G)$ be a vertex cover of $G$, and $I := V(G) \setminus X$. Any two distinct vertices $u, v \in I$ are in the same *equivalence class* $\mathcal{S} \subseteq I$ if and only if $u$ and $v$ are false twins. Note that there are at most $2^{|X|}$ distinct equivalence classes for the vertices in $I$ with respect to $X$.

**Lemma 8.** *Given a graph $G$ and a vertex cover $X \subseteq V(G)$ of $G$, for any $\mathcal{B}$ consisting of closed neighborhoods of $G$, it holds that $NCTD(\mathcal{B}) \leq 2^{|X|+1} + |X|$.*

**Reduction Rule 2.** *Let $G$ be a graph with a vertex cover $X \subseteq V(G)$ of $G$ and let $q = 2^{2^{|X|}+|X|}+1$. If $k < 2^{2|X|+1} + |X|$ and there exist $q + 2k + 1$ vertices in $I := V(G) \setminus X$ that are pairwise false twins whose closed neighborhoods are in $\mathcal{B}$, then delete one of them.*

**Reduction Rule 3.** *Given a graph $G$ and a vertex cover $X \subseteq V(G)$ of $G$, if there exists a pair $u, v \in V(G)$ of false twins in $I := V(G) \setminus X$ such that $N[u], N[v] \notin \mathcal{B}$, then delete $v$.*

The next lemma is the key to proving Reduction rule 2 is safe. Roughly, it says that if there are too many false twins in $I$ whose closed neighborhoods are in $\mathcal{B}$, then, by the pigeonhole principle, two of them will have almost "identical" teaching sets. This will facilitate the deletion of one of them.

**Lemma 9.** *Let $G$ be a graph with a vertex cover $X \subseteq V(G)$ of $G$, let $u_1, \ldots, u_\ell$ be a set of pairwise false twins in the equivalence class $Q$ of $I := V(G) \setminus X$ such that $N[u_1], \ldots, N[u_\ell] \in \mathcal{B}$, let $k \leq 2^{2|X|+1} + |X|$, and let $q = 2^{2^{|X|}+|X|} + 1$. For any NCTM $T$ of size $k$ for $\mathcal{B}$, if $\ell > q + 2k$, then there exist $i, j \in [\ell]$ such that $u_i \in T(N[u_i])$, $u_j \in T(N[u_j])$, $T(N[u_i]) \cap X = T(N[u_j]) \cap X$, $T(N[u_i]) \cap (Q \setminus \{u_i\}) = \emptyset$ if and only if $T(N[u_j]) \cap (Q \setminus \{u_j\}) = \emptyset$, and $T(N[u_i]) \cap \mathcal{S} = \emptyset$ if and only if $T(N[u_j]) \cap \mathcal{S} = \emptyset$ for any equivalence class $\mathcal{S} \neq Q$ of $I$ with respect to $X$.*

**Lemma 10.** *Reduction rule 2 is safe for* N-NCTD.

*Proof.* Let $Q \subseteq I$ be a set of $q + 2k + 1$ pairwise false twins, and let $T$ be an NCTM of size at most $k$ for $\mathcal{B}$. By Lemma 9, there exist $u, v \in Q$ such that $u \in T(N[u])$, $v \in T(N[v])$, $T(N[u]) \cap X = T(N[v]) \cap X$, $T(N[u]) \cap (Q \setminus \{u\}) = \emptyset$ if and only if $T(N[v]) \cap (Q \setminus \{v\}) = \emptyset$, and $T(N[u]) \cap \mathcal{S} = \emptyset$ if and only if $T(N[v]) \cap \mathcal{S} = \emptyset$ for any equivalence class $\mathcal{S} \neq Q$ of $I$ with respect to $X$. Select any vertex $w \in V(G) \setminus \{v\}$ such that $v \in T(N[w])$. We show that removing $v$ from $T(N[w])$ and adding $z \in Q$ to $T(N[w])$ maintains that $T$ is an NCTM of size at most $k$ for $\mathcal{B}$ in $G$. Specifically, if $u \notin T(N[w])$, then $z = u$. Otherwise, $z$ is any vertex in $Q \setminus \{u, v\}$ such that $z \notin T(N[w])$ (if $Q \setminus \{v\} \subseteq T(N[w])$, then no vertex is added to $T(N[w])$).

Let $T'$ be the teaching map for $\mathcal{B}$ obtained from $T$ by applying the above procedure for all $w \in V(G) \setminus \{v\}$ such that $v \in T(N[w])$. Note that $T'$ has size at most $k$. We show that $T'$ is an NCTM for $\mathcal{B}$. Toward a contradiction, suppose there exist $x, y \in V(G)$ such that $T'$ is not non-clashing for $N[x]$ and $N[y]$. Due to how $T'$ was obtained from $T$, and as $T$ is an NCTM for $\mathcal{B}$ while $T'$ is not, then $v$ is the only vertex in $T(N[x]) \cup T(N[y])$ distinguishing $N[x]$ and $N[y]$. So, $u \in T'(N[x]) \cup T'(N[y])$ and, w.l.o.g., $v \in N[x]$ and $v \notin N[y]$. We do a case analysis (see Fig. 5).

**Case 1:** $x = v$. Here, $v \in T'(N[x])$ distinguishes $N[x]$ and $N[y]$, a contradiction.

---

[6]A kernelization algorithm for a parameterized problem $\Pi$ with input $I$ and parameter $p$, runs in polynomial time and transforms an instance $(I, p)$ of $\Pi$ into an equivalent instance $(I', p')$ of $\Pi$ such that $|I'|, p' \leq f(p)$, for a computable function $f$. $\Pi$ is FPT parameterized by $p$ if and only if it admits a kernelization algorithm.

**Case 2:** $x \in N(v)$ and $y \neq u$. As $v \in N[x]$, $v \notin N[y]$, $u$ and $v$ are false twins, and $y \neq u$, then $u \in N[x]$ and $u \notin N[y]$. So, $u \in T'(N[x]) \cup T'(N[y])$ distinguishes $N[x]$ and $N[y]$, a contradiction.

**Case 3:** $x \in N(v)$ and $y = u$. As in Case 2, it must be that $u \in N[x]$. Since $v \in N[x]$, there exists $r \in (T(N[x]) \cup T(N[v])) \setminus \{v\}$ that distinguishes $N[x]$ and $N[v]$. We now study subcases.

**Case 3.1:** $r \in T(N[v]) \cap X$. Here, $r \in T'(N[u]) \cap X$ distinguishes $N[x]$ and $N[u]$, a contradiction.

**Case 3.2:** $r \in T(N[v]) \cap \mathcal{S}$ for an equivalence class $\mathcal{S}$ of $I$ with respect to $X$. If $r = u$, then $\mathcal{S} = Q$ and there exists $z \in (T'(N[u]) \cap Q) \setminus \{u, v\}$ that distinguishes $N[x]$ and $N[u]$, a contradiction. Otherwise, there is $r' \in (T'(N[u]) \cap \mathcal{S}) \setminus \{u, v\}$ that distinguishes $N[x]$ and $N[u]$, a contradiction.

**Case 3.3:** $r \in T(N[x])$. In this case, $r \in T'(N[x])$. If $r \neq u$, then $r \in T'(N[x])$ distinguishes $N[x]$ and $N[u]$, a contradiction. Otherwise, $r = u$. Since we have dealt with all the other cases, we can further assume that $r = u$ is the only vertex in $(T(N[x]) \cup T(N[v])) \setminus \{v\}$ that distinguishes $N[x]$ and $N[v]$. Due to this property of $r$ and the fact that $T$ satisfies the non-clashing condition for $N[x]$ and $N[u]$, either $v \in T(N[x])$ or some vertex $r' \in T(N[u]) \setminus \{u\}$ distinguishes $N[x]$ and $N[u]$. In the former case, as $u, v \in T(N[x])$, there exists $z \in (T'(N[x]) \cap Q) \setminus \{u, v\}$ that distinguishes $N[x]$ and $N[u]$, a contradiction. In the latter case, if $r' \neq v$, then $r'$ is also in $T'(N[u])$, and so, it distinguishes $N[x]$ and $N[u]$, a contradiction. Otherwise, $r' = v$, and hence, there exists $z \in (T'(N[x]) \cap Q) \setminus \{u, v\}$ that distinguishes $N[x]$ and $N[u]$, a contradiction.

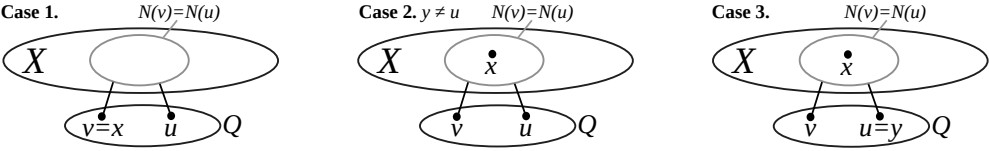

Figure 5: Cases 1, 2, and 3 in the forward direction of the proof of Lemma 10. Recall that $y \notin N[v]$.

That completes the case analysis, and thus, $T'$ is an NCTM for $\mathcal{B}$. Let $G' := G \setminus \{v\}$ and let $\mathcal{B}'$ equal $\mathcal{B}$ restricted to $G'$. Since the only teaching set of $T'$ containing $v$ is $T(N[v])$, then $T'$ restricted to $\mathcal{B}'$ is an NCTM of size at most $k$ for $\mathcal{B}'$ in $G'$. This completes the first direction of the proof.

For the reverse direction, let $T'$ be an NCTM of size at most $k$ for $\mathcal{B}'$ in $G'$. By Lemma 9, there exists $u \in V(G')$ such that $u \in T'(N[u])$ and $u$ and $v$ are false twins in $G$. Adding $v$ to $G'$ does not make any two closed neighborhoods that were the same in $G'$ become distinct in $G$. Hence, it is sufficient to extend $T'$ to an NCTM $T$ of size at most $k$ for $\mathcal{B}$ in $G$ as follows. Set $T(N[v]) := \{v\} \cup (T'(N[u]) \setminus \{u\})$, and, for all $x \in V(G')$, set $T(N[x]) := T'(N[x])$. Note that $T$ has size at most $k$. We now show that $T$ is an NCTM for $\mathcal{B}$ in $G$. As $T(N[x]) := T'(N[x])$ for all $x \in V(G')$, we only need to show that $T$ distinguishes $N[v]$ and $N[y]$ for all $y \in V(G) \setminus \{v\}$ such that $N[y] \in \mathcal{B}$. Since $v \in T(N[v])$, $T$ satisfies the non-clashing condition for $N[v]$ and $N[y]$ for any $y \notin N[v]$. Otherwise, $y \in X \cap N(v)$. If $x \in T'(N[u])$ distinguishes $N[u]$ and $N[y]$, then $x \neq u$ since $u \in N[y]$ as $u$ and $v$ are false twins, and thus, $x$ is also in $T(N[v])$ and distinguishes $N[v]$ and $N[y]$. Otherwise, $x \in T'(N[y])$ distinguishes $N[u]$ and $N[y]$, but $x \neq v$, and thus, $x \in T(N[y])$ also distinguishes $N[y]$ and $N[v]$. Hence, $T$ is an NCTM of size at most $k$ for $\mathcal{B}$ in $G$. $\quad\square$

**Lemma 11.** *Reduction rule 3 is safe for* N-NCTD.

*Proof of Theorem 7.* Use the classic polynomial-time algorithm to compute a 2-approximate minimum vertex cover $X \subseteq V(G)$. By Lemma 8, if $k \geq 2^{|X|+1} + |X|$, then it is a YES-instance. Otherwise, exhaustively apply Reduction rules 2 and 3, which do not alter $k$ and are safe by Lemmas 10 and 11. This is a kernelization algorithm as it takes polynomial time and the resultant graph has at most $2^{|X|}(q + 2k) + 2^{|X|} + |X| \leq 2^{|X|}(2^{2^{|X|}+|X|} + 2^{|X|+2} + 2|X|) + |X|$ vertices. $\quad\square$

## 4 COMBINATORIAL UPPER BOUNDS FOR GRAPH CLASSES

We finish by providing upper bounds on the (positive) non-clashing teaching dimension for closed neighborhoods $\mathcal{B}$ of a graph $G$ when $G$ is restricted to planar or unit square graphs. Note that upper bounds for any $\mathcal{B}$ hold for any $\mathcal{B}' \subseteq \mathcal{B}$, and $\text{VCD}(\mathcal{B}) \leq 4$ in both cases (Duraj et al., 2024).

**Theorem 12.** *For any planar graph $G$, if $\mathcal{B} = \{N[v] \mid v \in V(G)\}$, then $NCTD^+(\mathcal{B}) \leq 7$.*

*Proof.* We construct a positive NCTM $T$ of size at most 7 for $\mathcal{B}$. For all $v \in V(G)$, if $d(v) \leq 6$, then set $T(N[v]) := N[v]$. Otherwise, for each $v \in V(G)$ with $d(v) \geq 7$, place any 3 neighbors of $v$ in $T(N[v])$. Consider any one such vertex $v$ and let $v_1, v_2, v_3$ be its 3 neighbors in $T(N[v])$. Since $G$ is planar, there is at most one other vertex $u \in V(G)$ such that $v_1, v_2, v_3 \in N(u)$, as otherwise there would be a $K_{3,3}$. Hence, $T$ satisfies the non-clashing condition for $N[v]$ and $N[x]$ for all $x \in V(G) \setminus \{v, u, v_1, v_2, v_3\}$ if $u$ exists, and for all $x \in V(G) \setminus \{v, v_1, v_2, v_3\}$ otherwise.

As long as there exists a pair $N[v], N[y] \in \mathcal{B}$ such that $d(v) \geq 7$, $N[v] \neq N[y]$, and $T$ does not currently satisfy the non-clashing condition for the pair, then there exists $w \in (N[v] \setminus N[y]) \cup (N[y] \setminus N[v])$ that will be added to $T(N[v])$ or $T(N[y])$ to distinguish the pair. Specifically, if $w \in N[v] \setminus N[y]$, then add $w$ to $T(N[v])$, and otherwise, add $w \in N[y] \setminus N[v]$ to $T(N[y])$. In either case, $T$ then satisfies the non-clashing condition for $N[v]$ and $N[y]$. Once this process is completed, $T$ is a positive NCTM for $\mathcal{B}$ as it is non-clashing for $N[v]$ and $N[y]$ for any vertex $v$ with $d(v) \geq 7$, but also for the closed neighborhoods of any two vertices of degree at most 6 since their teaching sets coincide with their closed neighborhoods in this case. For each $v \in V(G)$ with $d(v) \geq 7$, as there were at most 4 possibilities for $y$ (the vertices corresponding to $u$, $v_1$, $v_2$, and $v_3$ above), then at most 4 vertices were added to any teaching set during this process. Thus, $T$ has size at most 7. $\quad\square$

**Theorem 13.** *For any planar graph $G$, if $\mathcal{B} = \{N[v] \mid v \in V(G)\}$, then $NCTD(\mathcal{B}) \leq 5$.*

We end with *unit square graphs*: the intersection graphs of axis-parallel unit squares in the plane.
**Theorem 14.** *For any unit square graph $G$, if $\mathcal{B} = \{N[v] \mid v \in V(G)\}$, then $NCTD^+(\mathcal{B}) \leq 4$.*

*Proof.* We construct a positive NCTM $T$ of size at most 4 for $\mathcal{B}$. For all $v \in V(G)$, let $R(v)$ be the minimum rectangle enclosing $N[v]$ in the plane (see Fig. 6). If a square $u$ does not intersect $v$, then for one of the directions left, right, up, or down, $u$ is farther in that direction than a square contained in $R(v)$ that is the farthest in that direction. Thus, $u$ is not contained in $R(v)$, and each square in $R(v)$ intersects $v$. For all $v \in V(G)$, $T(N[v])$ consists of the vertices represented by the leftmost, rightmost, topmost, and bottommost squares in $R(v)$ (they may not be unique). Consider any $u \in V(G) \setminus \{v\}$. If $R(u) \neq R(v)$, then there exists $x \in T(N[u]) \cup T(N[v])$ that distinguishes $N[u]$ and $N[v]$. Indeed, w.l.o.g., say that part of $R(u)$ is farther to the left than all of $R(v)$; then, $x \in T(N[u])$ is the leftmost square in $R(u)$, and $x \notin N[v]$ as part of $R(u)$ is farther to the left than all of $R(v)$. If $R(u) = R(v)$, then $N[u] = N[v]$ as each square in $R(v)$ ($R(u)$, resp.) intersects $v$ ($u$, resp.). $\quad\square$

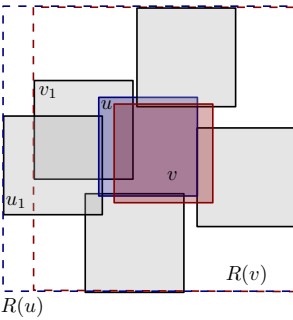

Figure 6: In the proof of Theorem 14, the minimum rectangles $R(u)$ (blue) and $R(v)$ (red) enclosing $N[u]$ and $N[v]$, respectively. The leftmost square in $R(u)$ ($u_1 \in T(N[u])$) is not contained in $R(v)$.

## 5   CONCLUSION

We studied non-clashing teaching for closed neighborhoods in graphs, notably providing highly non-trivial FPT algorithms and strong algorithmic lower bounds. Several open directions remain for future work, such as exploring the complexity with respect to treewidth or planarity for N-NCTD$^+$ and N-NCTD, as well as vertex integrity and treedepth for N-NCTD. However, like NON-CLASH (Ganian et al., 2025), we expect that they are W[1]-hard parameterized by the treewidth. It is also important to determine whether Theorem 13 is tight, since the balls of a planar graph has been mentioned (Chalopin et al., 2024) as a potential concept class $\mathcal{C}$ negatively answering the question of (Kirkpatrick et al., 2019; Fallat et al., 2023), i.e., such that $NCTD(\mathcal{C}) > VCD(\mathcal{C})$.

## ACKNOWLEDGMENTS

This work was partially supported by the Austrian Science Fund (FWF) [10.55776/Y1329 and 10.55776/COE12], the WWTF Vienna Science and Technology Fund (Project 10.47379/ICT22029), the ⬛ European Union's Horizon 2020 research and innovation COFUND programme Log-iCS@TUWien (grant agreement No 101034440), and the Smart Networks and Services Joint Undertaking (SNS JU) under the European Union's Horizon Europe and innovation programme under Grant Agreement No. 101139067 (ELASTIC). Views and opinions expressed are however those of the authors only and do not necessarily reflect those of the European Union (EU). Neither the EU nor the granting authority can be held responsible for them.

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

# A APPENDIX

## A.1 SUPPLEMENTARY FIGURE

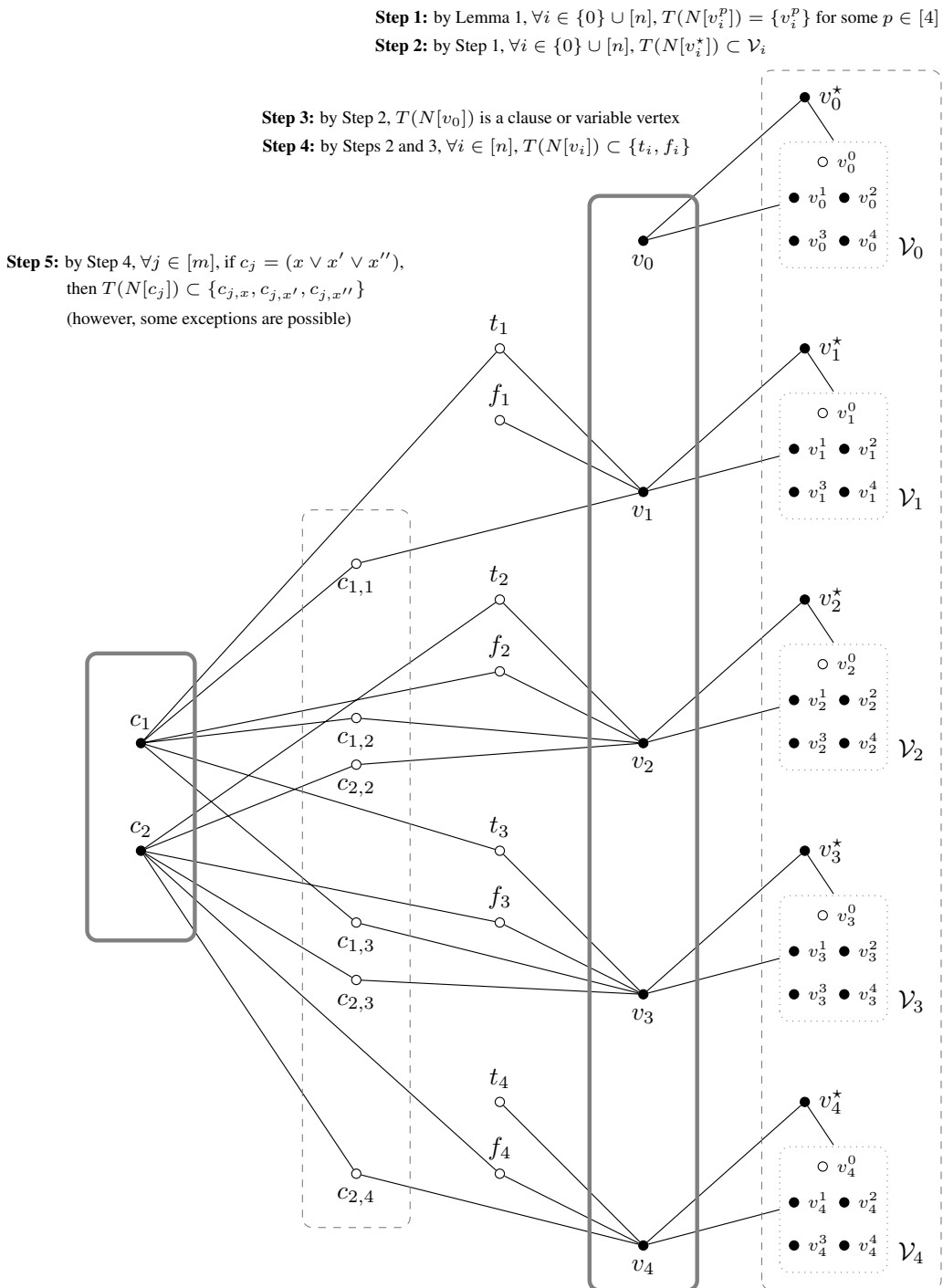

Figure 7: Example of the graph $G$ constructed in the proof of Theorem 2 with $\varphi = (\overline{x_1} \vee x_2 \vee \overline{x_3}) \wedge (\overline{x_2} \vee x_3 \vee x_4)$ in the 3-SAT instance. The two bold rectangles together form a clique. The edges going from vertices in one of the two large dashed rectangles to vertices in the other are omitted for legibility. Only the closed neighborhoods of filled vertices are in $\mathcal{B}$.

## A.2 Preliminaries

All considered graphs are connected and simple. Given a graph $G$ and $u \in V(G)$, $N(u) := \{v \mid uv \in E(G)\}$ is the *open neighborhood* of $u$, and $N[u] := N(u) \cup \{u\}$ is its *closed neighborhood*.

## A.3 Proof of Lemma 1

*Proof.* If no NCTM of size 1 for $\mathcal{B}$ exists, then the statement trivially holds. So, let $T$ be an NCTM of size 1 for $\mathcal{B}$. When only considering $N[u_1], \ldots, N[u_4]$ from $\mathcal{B}$, for any $i \in [4]$ and $v \in T(N[u_i])$,

- if $v = u_i$, then $v$ only distinguishes $N[u_i]$ and $N[u_j]$ for all $j \in [4] \setminus \{i\}$;

- if $v = u_j$ for some $j \in [4] \setminus \{i\}$, then $v$ only distinguishes $N[u_i]$ and $N[u_j]$;

- if $v \notin \{u_1, \ldots, u_4\}$, then $v$ does not distinguish $N[u_i]$ and $N[u_j]$ for any $j \in [4] \setminus \{i\}$.

Toward a contradiction, suppose that $T(N[u_i]) \neq \{u_i\}$ for all $i \in [4]$. By the above analysis, for all $i \in [4]$, the sole vertex in $T(N[u_i])$ distinguishes $N[u_i]$ and at most one other closed neighborhood in $\{N[u_1], \ldots, N[u_4]\} \setminus \{N[u_i]\}$. As there are $4(3)/2 = 6$ such pairs that need to be distinguished, $T$ does not satisfy the non-clashing condition for some pair in $\{N[u_1], \ldots, N[u_4]\}$, a contradiction. Thus, there exists $i \in [4]$ such that $T(N[u_i]) = \{u_i\}$. $\square$

## A.4 Proof of Theorem 3

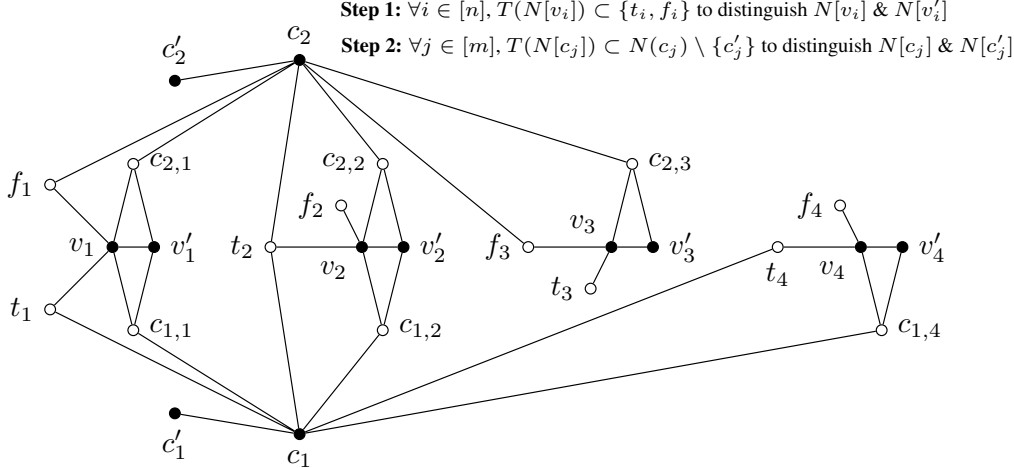

Figure 8: Example of the graph $G$ constructed in the proof of Theorem 3 with $\varphi = (\overline{x_1} \vee \overline{x_2} \vee \overline{x_4}) \wedge (x_1 \vee \overline{x_2} \vee x_3)$ in the 3-SAT instance. Only the closed neighborhoods of filled vertices are in $\mathcal{B}$.

*Proof.* Given an instance $\varphi$ of $n$-variable and $m$-clause 3-SAT, we construct a graph $G$ as follows:

1. for each clause $C_j$ in $\phi$, there is a clause vertex $c_j$ adjacent to a pendent vertex $c'_j$;

2. for each variable $x_i$ in $\phi$, there is a variable vertex $v_i$ adjacent to a vertex $v'_i$ as well as two literal vertices $t_i$ and $f_i$;

3. for all $i \in [n]$, if the literal $x_i$ ($\overline{x}_i$, resp.) is contained in the clause $C_j \in \phi$, then the literal vertex $f_i$ ($t_i$, resp.) is adjacent to the clause vertex $c_j$;

4. for all $i \in [n]$ and $j \in [m]$ such that $c_j$ is adjacent to $t_i$ or $f_i$, there is a vertex $c_{j,i}$ adjacent to $v_i$, $v'_i$, and $c_j$.

This concludes the construction of $G$ (see Fig. 8). We set $\mathcal{B}$ to contain the closed neighborhood of each variable vertex $v_i$, each clause vertex $c_j$, and each of their pendent vertices $v'_i$ and $c'_j$. So, from $\varphi$, the reduction outputs the N-NCTD$^+$ instance $(G, \mathcal{B}, 1)$. We now prove the reduction is correct.

Let $T$ be a positive NCTM of size 1 for $\mathcal{B}$. For all $i \in [n]$, $T(N[v_i]) \subset \{t_i, f_i\}$ as $N[v_i'] \subset N[v_i]$ and $N[v_i] \setminus N[v_i'] = \{t_i, f_i\}$. We extract a variable assignment $\tau \colon \{x_1, \ldots, x_n\} \to \{\texttt{True}, \texttt{False}\}$ as follows: if $T(N[v_i]) = \{t_i\}$, then $\tau(x_i) = \texttt{True}$, and otherwise, $\tau(x_i) = \texttt{False}$.

Toward a contradiction, suppose that $\tau$ is not a satisfying assignment for $\varphi$. Then, there exists a clause $C_j$ in $\varphi$ that is not satisfied by $\tau$. Let $x$, $x'$, and $x''$ be the 3 variables in $C_j$ in $\varphi$ (and the 3 corresponding variable vertices in $G$). Since $C_j$ is not satisfied, $T(N[x]), T(N[x']), T(N[x'']) \subset N[c_j]$ by the construction of $G$ and the definition of $\tau$. Thus, as $T$ is non-clashing for $N[c_j]$ and each of $N[x], N[x']$, and $N[x'']$, then $T(N[c_j])$ must consist of a vertex $u \in V(G)$ such that $u \in N[c_j]$, $u \notin N[x]$, $u \notin N[x']$, and $u \notin N[x'']$. It is easy to check that the only possibilities for such a vertex $u$ are $c_j$ and $c_j'$. However, $T(N[c_j]) \not\subset \{c_j, c_j'\}$ since $N[c_j'] = \{c_j, c_j'\} \subset N[c_j]$, as otherwise $T$ does not satisfy the non-clashing condition for $N[c_j]$ and $N[c_j']$. Hence, this contradicts that $T$ is an NCTM for $\mathcal{B}$. Therefore, $\tau$ is a satisfying assignment for $\varphi$.

For the reverse direction, let $\tau \colon \{x_1, \ldots, x_n\} \to \{\texttt{True}, \texttt{False}\}$ be a satisfying assignment of the 3-SAT formula $\varphi$. We define a positive teaching map $T$ for $\mathcal{B}$. For all $i \in [n]$, if $\tau(x_i) = \texttt{True}$, then $T(N[v_i]) := \{t_i\}$, and otherwise, $T(N[v_i]) := \{f_i\}$. For all $i \in [n]$, $T(N[v_i']) := \{v_i'\}$. For all $j \in [m]$, there exists $i \in [n]$ such that $\tau(x_i)$ satisfies $C_j$, and we set $T(N[c_j]) := \{c_{j,i}\}$. Lastly, for all $j \in [m]$, $T(N[c_j']) := \{c_j'\}$. We now prove that $T$ is a positive NCTM for $\mathcal{B}$.

For all $i \in [n]$, $T(N[v_i]) \cap N[v_i'] = \emptyset$, and thus, $T(N[v_i])$ distinguishes $N[v_i]$ and $N[v_i']$. Moreover, for all $i \in [n]$, $T(N[v_i']) = v_i'$ distinguishes $N[v_i']$ and all the other remaining closed neighborhoods in $\mathcal{B}$. For distinct $i, \ell \in [n]$, $T(N[v_i])$ distinguishes $N[v_i]$ and $N[v_\ell]$ since $T(N[v_i]) \cap N[v_\ell] = \emptyset$. By symmetric arguments, for each $j \in [m]$, $T$ satisfies the non-clashing condition for $N[c_j']$ and any other closed neighborhood in $\mathcal{B}$, as well as any pair of closed neighborhoods of clause vertices.

For all $i \in [n]$ and $j \in [m]$ such that $c_{j,i} \in T(N[c_j])$, $\tau(x_i)$ satisfies the clause $C_j$ in $\varphi$, and thus, $T(N[v_i]) \cap N[c_j] = \emptyset$, resulting in $N[v_i]$ and $N[c_j]$ being distinguished by $T(N[v_i])$. For all $i \in [n]$ and $j \in [m]$ such that $c_{j,i} \notin T(N[c_j])$, $N[v_i]$ and $N[c_j]$ are distinguished by $T(N[c_j])$. The case analysis is complete, and so, $T$ is a positive NCTM for $\mathcal{B}$.

Finally, note that both $|V(G)|$ and $|E(G)|$ are linear in the number of variables and clauses in the 3-SAT instance $\varphi$. So, if N-NCTD$^+$ can be solved in $2^{o(f(k) \cdot (|V(G)| + |E(G)|))}$ time for a computable function $f$, then 3-SAT can be solved in $2^{o(n+m)}$ time as $f(k) = \mathcal{O}(1)$, contradicting the ETH. $\quad\square$

### A.5  PROOF OF LEMMA 6

*Note that only the reverse direction in this proof differs from the proof sketch in the main text.*

*Proof.* First, we provide intuition. If $\ell$ is large, by the pigeonhole principle, there are 3 automorphic components in $A$ that are "identical" with respect to their adjacencies in $X$, and the closed neighborhoods of $\mathcal{B}$ that they contain along with their teaching sets. We select one of these 3 components, consider each teaching set that contains a vertex from it, and replace that vertex by one of its two automorphic copies (made possible by their identicalness), allowing that component to be deleted.

Formally, let $X \subseteq V(G)$ be such that $A = \{A_1, \ldots, A_\ell\}$ is a subset of the connected components of $G - X$, where $\max_{i \in [\ell]} |A_i| = t$ and $\ell > (|X| + t) \cdot 2^{(|X| + t)^2} \cdot 2^{2t + |X| + 1}$. Let $T$ be a positive NCTM of size at most $k$ for $\mathcal{B}$. We first prove that there exist $A_P, A_Q, A_R \in A$ that are automorphic to each other and such that, for any distinct $A_C, A_D \in \{A_P, A_Q, A_R\}$, there exists an automorphism $\sigma \colon V(A_C) \to V(A_D)$ where, for all $c \in V(A_C)$ and $d \in V(A_D)$ with $\sigma(c) = d$, it holds that:

**1**. $N(c) \cap X = N(d) \cap X$;

**2**. $N[c] \in \mathcal{B}$ if and only if $N[d] \in \mathcal{B}$;

**3**. if $N[c], N[d] \in \mathcal{B}$, then $T(N[c]) \cap X = T(N[d]) \cap X$ and, for all $u \in V(A_C)$ and $v \in V(A_D)$ with $\sigma(u) = v$, we have $u \in T(N[c]) \cap V(A_C)$ if and only if $v \in T(N[d]) \cap V(A_D)$.

We prove this via the pigeonhole principle. Trivially, the number of non-automorphic graphs on at most $|X| + t$ vertices is at most $(|X| + t) \cdot 2^{(|X| + t)^2}$. Thus, since $\ell > 2(|X| + t) \cdot 2^{(|X| + t)^2}$, there exist $A_P, A_Q, A_R \in A$ that are automorphic to each other and such that, for any distinct $A_C, A_D \in \{A_P, A_Q, A_R\}$, there exists an automorphism $\sigma \colon V(A_C) \to V(A_D)$ where, for all

$c \in V(A_C)$ and $d \in V(A_D)$ with $\sigma(c) = d$, 1. holds. Since $\ell > 2(|X| + t) \cdot 2^{(|X|+t)^2} \cdot 2^t$, we also have that 2. holds. Moreover, as $\ell > 2(|X| + t) \cdot 2^{(|X|+t)^2} \cdot 2^t \cdot 2^{t+|X|}$, we also have that 3. holds.

Select $w \in V(G) \setminus V(A_P)$ such that there exists $v \in T(N[w]) \cap V(A_P)$; we can assume $w$ exists, as otherwise we delete $V(A_P)$. Note that $w \in X$. We show that removing $v$ from $T(N[w])$ and adding a particular vertex $z \in V(A_Q) \cup V(A_R)$ to $T(N[w])$ maintains that $T$ is a positive NCTM for $\mathcal{B}$ in $G$. Let $\sigma_{P,Q} : V(A_P) \to V(A_Q)$ ($\sigma_{P,R} : V(A_P) \to V(A_R)$, resp.) be the automorphism between $V(A_P)$ and $V(A_Q)$ ($V(A_R)$, resp.). If $T(N[w]) \cap V(A_Q) = \emptyset$, then $z = \sigma_{P,Q}(v)$, and if not, but $\sigma_{P,R}(v) \notin T(N[w])$, then $z = \sigma_{P,R}(v)$, and otherwise no vertex is added to $T(N[w])$.

Let $T'$ be the teaching map for $\mathcal{B}$ obtained from $T$ by applying the above procedure for all $w \in X$ and $v \in V(A_P)$ such that $v \in T(N[w])$. Note that $T'$ has size at most $k$, and $T'$ is a positive teaching map for $\mathcal{B}$ since $T$ is, and due to property 1. above. We show that $T'$ is a positive NCTM for $\mathcal{B}$. Toward a contradiction, suppose there exist $x, y \in V(G)$ such that $T'$ does not satisfy the non-clashing condition for $N[x]$ and $N[y]$. Due to how $T'$ was obtained from $T$, and since $T$ is an NCTM for $\mathcal{B}$, while $T'$ is not, then the only vertices in $T(N[x]) \cup T(N[y])$ distinguishing $N[x]$ and $N[y]$ are in $V(A_P)$. Thus, w.l.o.g., $v \in (T(N[x]) \cup T(N[y])) \cap V(A_P)$ distinguished $N[x]$ and $N[y]$. Then, $\sigma_{P,Q}(v) \in T'(N[x]) \cup T'(N[y])$ and/or $\sigma_{P,R}(v) \in T'(N[x]) \cup T'(N[y])$, and, w.l.o.g., $v \in N[x]$ and $v \notin N[y]$. We do a case analysis (see Fig. 3). Let $u = \sigma_{P,Q}(v)$ and $u' = \sigma_{P,R}(v)$.

**Case 1:** $x \in N[v] \cap V(A_P)$. Thus, $v \in T'(N[x])$ distinguishes $N[x]$ and $N[y]$, a contradiction.

**Case 2:** $x \in N(v) \cap X$ and $y \notin N[u]$. As $N(v) \cap X = N(u) \cap X = N(u') \cap X$, then $u, u' \in N[x]$. Hence, $u', u$ or a vertex in $T'(N[x]) \cap V(A_Q)$ distinguishes $N[x]$ and $N[y]$, a contradiction.

**Case 3:** $x \in N(v) \cap X$ and $y \in N[u]$. Thus, $y \in V(A_Q)$. Let $\sigma_{P,Q}(y') = y$. As $T$ is an NCTM for $\mathcal{B}$, there exists $r \in T(N[x]) \cup T(N[y'])$ that distinguishes $N[x]$ and $N[y']$. We now study subcases.

**Case 3.1:** $r \in T(N[y'])$. Here, $r \in T'(N[y])$ distinguishes $N[x]$ and $N[y]$, a contradiction.

**Case 3.2:** $r \in T(N[x])$. If $r \notin V(A_P)$, then $r \in T'(N[x])$. If $r \in V(A_P)$, then $\sigma_{P,Q}(r) \in T'(N[x])$ and/or $\sigma_{P,R}(r) \in T'(N[x])$. We again study subcases.

**Case 3.2.1:** $r \notin N[y]$. If $r \notin V(A_P)$, then $r \in T'(N[x])$ distinguishes $N[x]$ and $N[y]$, a contradiction. Otherwise, as $r \in T(N[x])$ distinguishes $N[x]$ and $N[y']$, then $r \notin N[y']$, which implies that $\sigma_{P,Q}(r) \notin N[y]$. So, $\sigma_{P,Q}(r)$ or $\sigma_{P,R}(r)$ in $T'(N[x])$ distinguishes $N[x]$ and $N[y]$, a contradiction.

**Case 3.2.2:** $r \in N[y]$. As $T$ is an NCTM for $\mathcal{B}$, there exists $r' \in T(N[x]) \cup T(N[y])$ distinguishing $N[x]$ and $N[y]$. If $r' \in T(N[y])$, then $r' \in T'(N[y])$ distinguishes $N[x]$ and $N[y]$. Hence, $r' \in T(N[x])$. Further, if $r' \notin N[y']$, then either $r' \notin V(A_P)$, and so, $r' \in T'(N[x])$, or $r' \in V(A_P)$, in which case $\sigma_{P,R}(r') \in T'(N[x])$ as $r \in T'(N[x]) \cap N[y]$. In both cases, $T'$ satisfies the non-clashing condition for $N[x]$ and $N[y]$, a contradiction. Thus, $r' \in N[y']$, and as $r \in T'(N[x]) \cap N[y]$, then $\sigma_{P,R}(r') \in T'(N[x])$, which distinguishes $N[x]$ and $N[y]$, a contradiction.

That completes the case analysis, and thus, $T'$ is a positive NCTM for $\mathcal{B}$. Let $G' := G - A_P$ and let $\mathcal{B}'$ equal $\mathcal{B}$ restricted to $G'$. Since the only teaching sets of $T'$ containing vertices in $V(A_P)$ are those of the closed neighborhoods centered in vertices of $V(A_P)$, then $T'$ restricted to $\mathcal{B}'$ is a positive NCTM of size at most $k$ for $\mathcal{B}'$ in $G'$. This completes the first direction of the proof.

For the reverse direction, we do the (simpler) inverse of the above. Let $T'$ be a positive NCTM of size at most $k$ for $\mathcal{B}'$ in $G'$. Since there exists an automorphism $\sigma_{Q,R} : V(A_Q) \to V(A_R)$, then, for each $q \in V(A_q)$ such that $N[q] \in \mathcal{B}'$, $T'(N[q]) \cap V(A_Q) \neq \emptyset$ and $T'(N[\sigma_{Q,R}(q)]) \cap V(A_R) \neq \emptyset$. Indeed, otherwise $T'$ does not satisfy the non-clashing condition for that pair of closed neighborhoods. Note that the addition of $V(A_P)$ to $G'$ does not make any two closed neighborhoods that were the same in $G'$ become distinct in $G$. Hence, it is sufficient to extend $T'$ to a positive NCTM $T$ of size at most $k$ for $\mathcal{B}$ in $G$ as follows. For all $p \in V(A_P)$, let $p' := \sigma_{P,Q}(p)$. For all $x \in V(G')$, set $T(N[x]) := T'(N[x])$, and, for all $p \in V(A_P)$ such that $N[p] \in \mathcal{B}$, set $T(N[p]) := (T(N[p']) \cap X) \bigcup_{q \in T(N[p']) \cap V(A_Q)} \{\sigma_{P,Q}^{-1}(q)\}$.

Note that $T$ has size at most $k$ and is a positive teaching map for $\mathcal{B}$. We now show that $T$ is a positive NCTM for $\mathcal{B}$ in $G$. Since $T(N[x]) := T'(N[x])$ for all $x \in V(G')$, we only need to show that $T$ distinguishes $N[p]$ and $N[y]$ for all $p \in V(A_P)$ and $y \in V(G)$ such that $N[p], N[y] \in \mathcal{B}$. For all $p \in V(A_P)$, since $T(N[p]) \cap V(A_P) \neq \emptyset$, $T$ satisfies the non-clashing condition for $N[p]$ and $N[y]$ for any $y \notin (V(A_P) \cup (N(p) \cap X))$. We now proceed with a case analysis.

**Case 1:** $y \in V(A_P) \setminus \{p\}$. Let $x \in T'(N[\sigma_{P,Q}(y)]) \cup T'(N[p'])$ distinguish $N[\sigma_{P,Q}(y)]$ and $N[p']$. If $x \in X$, then $x \in T(N[p]) \cup T(N[y])$ distinguishes $N[p]$ and $N[y]$. Otherwise, $x \in V(A_Q)$, and thus, $\sigma_{P,Q}^{-1}(x) \in T(N[p]) \cup T(N[y])$ distinguishes $N[p]$ and $N[y]$.

**Case 2:** $y \in N(p) \cap X$. If $x \in T'(N[p'])$ distinguishes $N[p']$ and $N[y]$, then either $x \in T(N[p]) \cap X$ or $x \in V(A_Q)$ and $\sigma_{P,Q}^{-1}(x) \in T(N[p])$. In both cases, $T$ satisfies the non-clashing condition for $N[p]$ and $N[y]$. Otherwise, $x \in T'(N[y])$ distinguishes $N[p']$ and $N[y]$. Since $x \notin N[p']$ and $T'(N[y]) \cap V(A_P) = \emptyset$, then $x \notin N[p]$, and thus, $x \in T(N[y])$ distinguishes $N[p]$ and $N[y]$.

This completes the case analysis. Hence, $T$ is a positive NCTM of size at most $k$ for $\mathcal{B}$ in $G$. □

### A.6 PROOF OF LEMMA 8

*Proof.* It suffices to prove it for $\mathcal{B} = \{N[v] \mid v \in V(G)\}$ as the upper bound then holds for all $\mathcal{B}' \subseteq \mathcal{B}$. We construct an NCTM $T$ of size at most $2^{|X|+1} + |X|$ for $\mathcal{B}$. Let $\mathcal{S}_1, \ldots, \mathcal{S}_p$ be the distinct equivalence classes of size at least 1 of $I := V(G) \setminus X$ with respect to $X$. For each $x \in X$, set $T(N[x]) := X \cup \{s_1, \ldots, s_p\} \cup \{t_1, \ldots, t_p\}$, where $s_i$ and $t_i$ are any two distinct vertices in $\mathcal{S}_i$ for all $i \in [p]$ (if $|S_i| = 1$, then say that $t_i$ does not exist). For each $y \in I$, set $T(N[y]) := X \cup \{y\}$. Since $p \leq 2^{|X|}$, $T$ has size at most $2^{|X|+1} + |X|$. We now prove that $T$ is an NCTM for $\mathcal{B}$. Consider the pair $N[u]$ and $N[v]$ for any two distinct vertices $u, v \in V(G)$. If $u, v \in I$, then $T$ is non-clashing for $N[u]$ and $N[v]$ as $y \in T(N[y])$ for all $y \in I$. If $u \in X$ and $v \in I$, then $T$ is non-clashing for $N[u]$ and $N[v]$ since if $N[u] \neq N[v]$, then either $v \in T(N[v])$ or some vertex in $X$ distinguishes $N[u]$ and $N[v]$, or some vertex in $T(N[u]) \cap (I \setminus \{v\})$ distinguishes them. Lastly, if $u, v \in X$ and $N[u] \neq N[v]$, then some vertex in $X$ is only in one of $N[u]$ and $N[v]$ and/or there exists an equivalence class $\mathcal{S}$ of $I$ such that all of $\mathcal{S}$ is in $N[u]$ or $N[v]$ and none of $\mathcal{S}$ is in the other. In the first case, $T$ is non-clashing for $N[u]$ and $N[v]$ as $X \subseteq T(N[x])$ for all $x \in X$. In the second case, $T$ is non-clashing for $N[u]$ and $N[v]$ as $\{s_1, \ldots, s_p\} \subseteq T(N[x])$ for all $x \in X$. □

### A.7 PROOF OF LEMMA 9

*Proof.* First, we prove that, for any NCTM $T$ of size $k$ for $\mathcal{B}$, if there exist $q$ distinct indices $p \in [\ell]$ such that $u_p \in T(N[u_p])$, then there exist distinct $i, j \in [\ell]$ such that $u_i \in T(N[u_i])$, $u_j \in T(N[u_j])$, $T(N[u_i]) \cap X = T(N[u_j]) \cap X$, $T(N[u_i]) \cap (Q \setminus \{u_i\}) = \emptyset$ if and only if $T(N[u_j]) \cap (Q \setminus \{u_j\}) = \emptyset$, and $T(N[u_i]) \cap \mathcal{S} = \emptyset$ if and only if $T(N[u_j]) \cap \mathcal{S} = \emptyset$ for any equivalence class $\mathcal{S} \neq Q$ of $I$ with respect to $X$. So, assume that there exist $q$ distinct indices $p \in [\ell]$ such that $u_p \in T(N[u_i])$. Since there are at most $2^{|X|}$ equivalence classes of $I$ with respect to $X$, there are at most $2^{2^{|X|}}$ distinct subsets of these equivalence classes. Hence, there are at most $2^{2^{|X|}+|X|}$ distinct subsets of $X$ and the equivalence classes. Thus, the existence of $u_i$ and $u_j$ with the above properties follows by the pigeonhole principle since $q = 2^{2^{|X|}+|X|} + 1 > 2^{2^{|X|}+|X|}$.

It remains to prove that, for any NCMT $T$ of size $k$ for $\mathcal{B}$, if $\ell > q + 2k$, then there exist $q$ distinct indices $p \in [\ell]$ such that $u_p \in T(N[u_p])$. Toward a contradiction, let $T$ be an NCTM of size $k$ for $\mathcal{B}$, let $\ell > q + 2k$, and suppose there are at most $q - 1$ distinct indices $p \in [\ell]$ such that $u_p \in T(N[u_p])$. For all $i \in [\ell]$, if $u_i \notin T(N[u_i])$, then each of the at most $k$ vertices in $T(N[u_i])$ distinguishes $N[u_i]$ and at most one other closed neighborhood in $\{N[u_1], \ldots, N[u_\ell]\}$. Thus, w.l.o.g., for all $i \in [q - 1]$, let $u_i \in T(N[u_i])$. By the above analysis, the teaching sets of $N[u_q], \ldots, N[u_\ell]$ combined distinguish at most $(\ell - q + 1)k$ pairs in $\{N[u_q], \ldots, N[u_\ell]\}$. Note that $(\ell - q + 1)(\ell - q)/2$ such pairs must be distinguished and

$$(\ell - q + 1)(\ell - q)/2 > (\ell - q + 1)k \iff (\ell - q)/2 > k \iff \ell > q + 2k.$$

Thus, since $\ell > q + 2k$, $T$ does not distinguish a pair in $\{N[u_q], \ldots, N[u_\ell]\}$, a contradiction. □

### A.8 PROOF OF LEMMA 11

*Proof.* Let $T$ be an NCTM of size at most $k$ for $\mathcal{B}$. For all $w \in V(G) \setminus \{v\}$ such that $v \in T(N[w])$, remove $v$ from $T(N[w])$, and if $u \notin T(N[w])$, then add $u$ to $T(N[w])$. From the proof that Reduction rule 2 is safe (Lemma 10), it is easy to see that this maintains that $T$ is an NCTM of size at most $k$ for $\mathcal{B}$ since $N[u]$ and $N[v]$ are not in $\mathcal{B}$. The reverse direction is trivial since any NCTM for $\mathcal{B}'$ in $G'$ is also an NCTM for $\mathcal{B}$ in $G$ as $N[u]$ and $N[v]$ are not in $\mathcal{B}$. □

### A.9 Proof of Theorem 13

*Proof.* We construct an NCTM $T$ of size at most 5 for $\mathcal{B}$. For all $v \in V(G)$, if $d(v) \leq 4$, then set $T(N[v]) := N[v]$. Otherwise, for each $v \in V(G)$ with $d(v) \geq 5$, place any 3 neighbors of $v$ in $T(N[v])$. Consider any one such vertex $v$ and let $v_1, v_2, v_3$ be its 3 neighbors in $T(N[v])$. Then, as in the proof of Theorem 12, there is at most one other vertex $u \in V(G)$ such that $v_1, v_2, v_3 \in N(u)$, and hence, $T$ satisfies the non-clashing condition for $N[v]$ and $N[x]$ for all $x \in V(G) \setminus \{v, u, v_1, v_2, v_3\}$ if $u$ exists, and for all $x \in V(G) \setminus \{v, v_1, v_2, v_3\}$ otherwise. If $u \notin N[v]$, then add $u$ to $T(N[v])$, and note that this ensures that $T$ satisfies the non-clashing condition for $N[v]$ and $N[z]$ for all $z \in V(G) \setminus \{v\}$ since $u \notin N[v]$, but $u \in N[v_1]$, $u \in N[v_2]$, and $u \in N[v_3]$. Hence, in what follows, if $u$ exists, then assume that $u \in N[v]$.

As long as there exists a pair $N[v], N[y] \in \mathcal{B}$ such that $d(v) \geq 5$, $N[v] \neq N[y]$, and $T$ does not currently satisfy the non-clashing condition for the pair, then there exists $w \in (N[v] \setminus N[y]) \cup (N[y] \setminus N[v])$ that will be added to $T(N[v])$ or $T(N[y])$ to distinguish the pair. Specifically, if $w \in N[v] \setminus N[y]$, then add $w$ to $T(N[v])$, and otherwise, add $w \in N[y] \setminus N[v]$ to $T(N[y])$. In either case, $T$ then satisfies the non-clashing condition for $N[v]$ and $N[y]$. Once this process is completed, $T$ is an NCTM for $\mathcal{B}$ as it is non-clashing for $N[v]$ and $N[y]$ for any vertex $v$ with $d(v) \geq 5$, but also for the closed neighborhoods of any two vertices of degree at most 4 since their teaching sets coincide with their closed neighborhoods in this case. We now argue that there were at most 2 possibilities for $y$ above. The vertices in $\{u, v, v_1, v_2, v_3\}$ cannot form a $K_5$ since $G$ is planar. Thus, one of the edges $v_1 v_2$, $v_1 v_3$, and $v_2 v_3$ is not in $E(G)$ since $uv \in E(G)$. Say, w.l.o.g., that $v_2 v_3 \notin E(G)$. Then, $y$ could only have been $u$ or $v_1$. Hence, at most 2 vertices were added to any teaching set during the above process, and thus, $T$ has size at most 5. □

