# OpenReview forum: "Non-Clashing Teaching in Graphs: Algorithms, Complexity, and Bounds"
_ICLR.cc/2026/Conference — ICLR 2026 Poster_

### Official Review · Reviewer_G64H · 2025-10-24

**Soundness:** 3
**Presentation:** 2
**Contribution:** 3
**Rating:** 4
**Confidence:** 3

**Summary:**

This paper considers computational complexity aspects of a generalization of the Non-Clashing Teaching Dimension (NCTD) of concept classes. The basic definition of the NCTD, introduced and analyzed in previous works, is as follows. There is a finite binary concept class $C$ and a (computationally powerful) teacher $T$. The teacher maps each $C$ to a set of labeled examples $T(C)$ representing $C$. The constraint is that for each $C \neq C'$, we should have an example in $T(C) \cup T(C')$ that is consistent with only one of the concepts. There is also a version NCTD+ that only allows positive labels.

It turns out that the concept of NCTD can be equivalently represented using balls in graphs. The current work focuses on a more specific case, where the balls are only of radius 1 ("closed neighborhood"). The authors prove a range of theoretical results, including an improved exponential conditional hardness (lower bound assuming ETH holds); near-matching upper bounds; Fixed Parameter Tractable upper bounds, parameterized by the tree-depth (among others); and NCTD upper bounds for specific graph classes, such as planar graphs.

**Strengths:**

- Strong theoretical results which extend the state of the art understanding of problems related to machine teaching.

- The proofs are interesting and well-written. I did not fully verify, but the main arguments seem believable.

**Weaknesses:**

- It is not clear what are the applications of the results, and how they relate to learning. If such applications exist, the author should make a better effort clarifying these.

- The paper is purely theoretical and might not be a best fit to a conference like ICLR. While the results seem interesting (even without concrete applications), they are probably a better fit in a more theoretical venue.

**Questions:**

I do not fully understand what is the connection between the result and the original definition of NCTD (for balls). While I understand that the problems are connected, I do not understand what exactly your results imply. Do they have interesting applications?

Do results for restricted graph classes have, e.g., graphs with low tree depth or planar graphs, have interesting applications? What is the typical structure of graphs in this setting?

---

> ### Author Response · Authors · 2025-11-21
>
> Thank you very much for your insightful review and interesting questions!
>
> General comments:
>
> 1. Apart from the applications mentioned mainly in the introduction, non-clashing teaching can be seen as an easier setting to study sample compression schemes, which constitute a very well-studied learning paradigm that we also mention in line 37. The main question there is whether every finite binary concept class $\mathcal{C}$ admits a sample compression scheme of size $O(VCD(\mathcal{C}))$, and the best result in general is still exponential in $VCD(\mathcal{C})$. In contrast to sample compression, here the teacher has full access to the domain when designing the teaching sets. However, despite this, we still do not know if NCTD($\mathcal{C}$) is $O(VCD(\mathcal{C}))$ for every finite binary concept class $\mathcal{C}$. Thus, studying NCTD can be seen as a stepping stone toward the sample compression conjecture. As one would also naturally be interested in computing the teaching sets, our algorithmic analysis is also motivated by this.
>
> 2. Learning theory is specifically mentioned as a relevant topic in the call for papers, and so naturally, many papers in this area will be purely theoretical, as was the case for many past ICLR papers. We also believe that other theoreticians outside of learning theory would also be interested in our paper. Lastly, our paper is also highly motivated by an ICLR 2025 paper as mentioned in the abstract and introduction. Thus, we feel that we do not fall outside of the scope of ICLR.
>
> Questions:
>
> 1. As discussed in lines 108-114, any finite binary concept class can be represented by a set of balls of radius 1 in a graph $G$, i.e., a set of closed neighborhoods in a graph $G$. As we show in our paper, in non-clashing teaching for closed neighborhoods, we are able to obtain tractability results that have proven elusive for non-clashing teaching for balls in graphs. These results are nicely summarized in Table 1. For instance, we have FPT algorithms w.r.t. treedepth for NCTD$^+$ and w.r.t. vertex cover for NCTD, both of which the status is unknown for balls in graphs. We also have much better bounds for planar graphs and unit square graphs. Finally, our lower bounds also apply to the case of balls in graphs, and we improve over the previous ones from the literature for both NCTD and NCTD$^+$. Overall, this implies that the representation of finite binary concept classes by closed neighborhoods in graphs is preferable over that of balls in graphs.
>
> 2. Our study focuses on the concept class of closed neighborhoods in graphs itself rather than how finite binary concept classes may look when represented by closed neighborhoods in graphs. This is motivated by the previous works on non-clashing teaching for balls in graphs, as well as all the works that have considered closed neighborhoods in graphs as concept classes (see lines 113-114). Despite this, we would find it interesting to know whether there are some other well-known concept classes that can be represented by a set of closed neighborhoods of a graph in some restricted graph class, but we are unaware of any. We highlight that there is a general trend toward studying concept classes derived from graph structures, with a long list of such papers provided in "F. Foucaud, H. Gahlawat, F. Mc Inerney, and P. Tale: The Parameterized Complexity of Computing the VC-Dimension", which is set to appear at NeurIPS 2025 and is available on arxiv (see the top of page 3 for the list).

---

### Official Review · Reviewer_HDoS · 2025-10-31

**Soundness:** 3
**Presentation:** 3
**Contribution:** 3
**Rating:** 6
**Confidence:** 3

**Summary:**

This work studies (positive) non-clashing teaching for closed neighborhoods, showing this concept class is broadly representative and yielding stronger ETH-based lower bounds and tighter algorithms than prior results for balls in graphs. It proves no $2^{o(f(k)·|V|)}$ algorithm for N-NCTD under ETH and gives a tight $2^{O(|E|)}$ algorithm with matching lower bound for N-NCTD+, nearly matching known upper bounds in the general case. It further shows FPT results under treedepth (positive variant) and vertex cover (general variant) and derives small combinatorial upper bounds for planar and unit-square graphs, clarifying when structure enables efficient teaching maps.

**Strengths:**

1. Near-tight complexity results, improving lower bounds and adding new FPT regimes.
2. Careful comparison to balls in graphs, showing where closed neighborhoods help.
3. Combinatorial upper bounds tied to structural classes and VC-dimension observations.

**Weaknesses:**

1. Exposition can be heavy; more intuition and illustrative examples would aid accessibility.
2. Limited discussion of practical or empirical implications for machine teaching applications.
3. ETH tightness is strong theoretically, but could be complemented by empirical hardness studies.

**Questions:**

1. Are constructive or approximate teaching strategies possible for additional graph families ?
2. What are realistic constants in the FPT algorithms and prospects for practical solvers?
3. A concise taxonomy contrasting closed neighborhoods vs.\ balls would be valuable.

---

> ### Author Response · Authors · 2025-11-21
>
> Thank you very much for your kind review and helpful comments!
>
> General comments:
>
> 1. To improve the exposition, we have added a figure for the equivalence between balls in graphs and binary concept classes, figures for the case analyses in the proofs of Lemmas 6 and 10, a figure illustrating the pruning of the treedepth decomposition for Theorem 5, and now Table 1, the figure illustrating a treedepth decomposition, and the figure for Theorem 14 are included in the main text. We have also added guided steps in both of the reduction figures (for Theorems 2 and 3) in the appendix to help the reader better understand the proofs just from the figures themselves. Further, we have simplified the statement (and thus proof) of Lemma 1 so that now it only covers our exact use of it. We have also made some minor adjustments to some of the other proofs in cases where we thought the reader could be aided a bit more.
>
> 2. and 3. As with many other past ICLR papers, our focus in this paper is purely theoretical, and given the number and complexity of the results, we feel that practical experiments are out of the scope. We have discussed some practical applications in the introduction, but in general we feel that an empirical analysis is most effective once the theoretical analysis is absolutely complete. Our paper moves us even closer to fully understanding where the boundary of tractability lies for these problems.
>
> Questions:
>
> 1. As mentioned in lines 86-87, constructive teaching strategies are known for trees, cycles, cacti, and interval graphs. We have pushed this a bit further with our results for planar and unit square graphs. It would be interesting if larger graph classes could be tackled in the same way, but already finding the optimal bounds for planar graphs is a significant challenge.
>
> 2. The constants for the FPT algorithm w.r.t. vertex cover are relatively small (the largest would be a multiplicative factor of 2 in the exponent that would come from applying the algorithm from Theorem 4). For the FPT algorithm w.r.t. treedepth, they would also not be too large, but this algorithm is impractical due to the function of treedepth being a tower of exponents. This result's importance is rather in demonstrating that an FPT algorithm parameterized by the treedepth is possible, which can now be followed up by trying to find the fastest such algorithm.
>
> 3. This taxonomy is provided in Table 1, which used to be in the appendix, but now features in the main text.

---

### Official Review · Reviewer_othc · 2025-11-01

**Soundness:** 4
**Presentation:** 3
**Contribution:** 2
**Rating:** 4
**Confidence:** 3

**Summary:**

This paper studies the notion of the *non-clashing teaching dimension* (NCTD) within the broader framework of machine teaching, focusing on both variants of non-clashing teaching for *balls in simple finite graphs with closed neighborhoods*.
Relying on the Exponential Time Hypothesis (ETH), the authors employ 3-SAT style reductions to establish hardness results and improve algorithmic bounds for the positive variant (**N-NCTD⁺**), while also providing a corresponding lower bound for **N-NCTD**.
They further design a fixed-parameter tractable (FPT) algorithm parameterized by *treedepth* for computing **N-NCTD⁺**, achieved via a pruning mechanism on the treedepth decomposition followed by a brute-force step that is fixed-parameter tractable in the treedepth parameter.
Additionally, they propose a set of safe reduction rules that yield an FPT algorithm parameterized by the *vertex cover number*.
The paper concludes with combinatorial arguments proving constant upper bounds on **N-NCTD⁺** for planar graphs with closed neighborhoods.

**Strengths:**

### Strengths

The lower bound improvement for **N-NCTD** is significant, and the **FPT parameterization by vertex cover** for **N-NCTD** is highly relevant.

I found the presentation style particularly effective, as several proofs are preceded by intuitive explanations and supported by neat, well-designed diagrams. This approach makes the paper engaging and easy to follow. I believe this style should be extended to other proofs as well, with additional illustrative examples where appropriate. For example, while showing the correctness of the reduction in **Theorem 2**, since the size of the **NCTM** is 1, an explicit map could be shown to help readers follow the proof without compromising generality.

I also appreciated the **combinatorial proofs for planar graphs**, which present a large use case and offer elegant solutions. However, the explanation could be made more verbose for better clarity. For instance, in the case where *d(v) ≥ 7*, it would help to explain why there are only four possibilities. A short intuitive note could clarify that this clashing arises only when all three choices match, or due to earlier shifting where two old and one new match occur—such combinations being \( \binom{3}{3} + \binom{3}{2} = 4 \). Including such brief intuitive explanations would greatly improve readability and pedagogical value.

**Weaknesses:**

### Weaknesses

The proofs and overall write-up are very specific and may primarily appeal to readers from the learning theory subcommunity. The paper is difficult to parse and, more importantly, to appreciate for readers without prior background in machine teaching or the concept of the non-clashing teaching dimension (NCTD).

Although the proofs presented in Section 2 appear to be correct and I verified all of them rigorously, I found the graph construction to be very similar to the one used in *“The Computational Complexity of Positive Non-Clashing Teaching in Graphs.”* While correctness is not in question, I believe it is important for proofs to introduce novel ideas or techniques that could be applicable in other contexts; I did not find such novelty in the proofs of this paper.

**Questions:**

### Practical Applications and Discussion

Can you list down the practical applications of these notions and improvements?
For **NCTD⁺**, how is it useful to use **treedepth** as compared to **vertex integrity** and what are the tangible benefits, in terms of theory and implementation perspectives?
Similar to **NCTD**, would it be beneficial to have **vertex cover** as it is very general and has good approximation bounds?
Can there be some practical experiments to quantify the impacts of **reduction rules** for FPT algorithms for **NCTD** and **NCTD⁺**?

---

> ### Author Response · Authors · 2025-11-21
>
> Thank you very much for your nice review and useful suggestions! We appreciate that you particularly liked our presentation style.
>
> As per your suggestions, we have added guided steps in both of the reduction figures (for Theorems 2 and 3) in the appendix to help the reader better understand the proofs just from the figures themselves. We have also added figures for the case analyses in the proofs of Lemmas 6 and 10, a figure illustrating the pruning of the treedepth decomposition for Theorem 5, and now the figure illustrating a treedepth decomposition and the figure for Theorem 14 are in the main text. Further, we have simplified the statement (and thus proof) of Lemma 1 so that now it only covers our exact use of it. We have also made some minor adjustments to some of the other proofs in cases where we thought the reader could be aided a bit more.
>
> Concerning the planar graphs proof, the four possibilities for $y$ are actually much simpler. They are just in fact the four cases not covered by the first paragraph: $u$, $v_1$, $v_2$, $v_3$. We have made the necessary adjustment to make this clearer.
>
> Regarding our paper primarily appealing to the learning theory subcommunity, we feel that this is the case for most papers and their respective domains at these conferences. While other subcommunities may be larger, learning theory is specifically mentioned as a relevant topic in the call for papers, and we believe that other theoreticians would also be interested in our paper.
>
> Concerning the graph constructions in Section 2 resembling the one used in “The Computational Complexity of Positive Non-Clashing Teaching in Graphs”: first, we will say that most reductions from 3-SAT will have some resemblance due to them all being reductions from the same problem. However, there are significant differences between ours and the one from that other paper. For N-NCTD$^+$, it is essential in our constructed graph $G$ to have both $|V(G)|$ and $|E(G)|$ be linear in the number of variables and clauses in the 3-SAT instance, which is not the case in the other paper. Also, our proof holds for $k=1$, while the one in the other paper only holds for $k=2$, which also induces differences in the constructions. For N-NCTD, the differences are much more significant as negatively-labeled examples are now allowed in our construction, while they are not in the other paper. This requires the introduction of the gadget $\mathcal{V}_i$ (which makes use of Lemma 1) with $v_i^*$, as well as $v_0$ in order to force many teaching sets in a certain way. This can be achieved much more simply in the positive case as in our reduction for N-NCTD$^+$ or in the other paper.
>
> Questions:
>
> 1. the practical applications have been mentioned in the first and fourth paragraphs of our paper. The improvements we have made are mainly algorithmic, precising the boundary of tractability for these problems. As we have made more algorithmic advances by considering closed neighborhoods in graphs than balls in graphs, we have also demonstrated that this concept class is more preferable given that it also has the equivalence with binary concept classes.
>
> 2. As mentioned in footnote 6, an FPT algorithm parameterized by the treedepth of the input graph implies an FPT algorithm parameterized by the vertex integrity of the input graph, and thus, the former is a stronger result than the other in terms of theory. In terms of implementation perspectives, now knowing that there exists an FPT algorithm parameterized by the treedepth, the next step is to either try to improve this again by demonstrating an FPT algorithm for a more general parameter, or to find the fastest such algorithm. Right now, as in the other papers in the literature, the focus is still on determining the theoretical limits of tractability before moving on to optimizing the practical implementation, as is typically done for all problems.
>
> 3. The answer is the same as the end of the answer to question 2 above. In general, the aim is always to find the most general parameter for which an FPT algorithm exists, and then to find the fastest such FPT algorithm.
>
> 4. As with many other past ICLR papers, our focus in this paper is purely theoretical, and given the number and complexity of the results, we feel that practical experiments are out of the scope. However, the reduction rules for the FPT algorithm for N-NCTD parameterized by the vertex cover number run in polynomial time (in fact, even quadratic time), and thus, are very efficient. On the other hand, the reduction rule for the FPT algorithm for N-NCTD$^+$ parameterized by the treedepth only runs in FPT time, but this runtime is far smaller than the runtime of the entire FPT algorithm.

---

> > ### Comment · Reviewer_othc · 2025-11-26
> >
> > I would like to thank the authors for addressing most of my major concerns. I do agree that multiple 3SAT reductions can have similar styles and appreciate novelties highlighted. I also really appreciate adding diagrams, following the proofs is much easier now! Having said that, I am still not completely sure if the proofs introduced( although correct and valuable), are inspiring new directions. Hence I would urge other reviewers and AC to carefully analyse this aspect as I am not an expert in this domain.
> >
> > But I certainly feel that the latest presentation style enables the proofs to reach a wider audience and hence raise my score.
> > Minor remark:
> >
> > To differentiate the additions/ modifications made during rebuttal, kindly use a different colored text

---

> > > ### Author Response · Authors · 2025-11-28
> > >
> > > Thank you very much! We are happy that you appreciate the latest presentation style.
> > >
> > > We will upload a new version highlighting the modifications with color ASAP.

---

### Official Review · Reviewer_BjtC · 2025-11-06

**Soundness:** 3
**Presentation:** 1
**Contribution:** 3
**Rating:** 4
**Confidence:** 3

**Summary:**

The paper studies non-clashing teaching paradigm in learning theory where a teacher can give the learner catered examples instead of random samples in a non-collusive way. Specifically, the authors look at the specific cases where the concept class comes from a closed neighborhood of a graph and provide many theoretical analyses of both upper and lower bounds for this particular problem (which is as general is binary concept class learning).

**Strengths:**

The theoretical contribution is significant, as far as the reviewer can tell studying the immediate literature. And the math is correct, as far as the reviewer has checked (barring cited results). While the reviewer is not an expert in the immediate literature and cannot speak to how challenging it is to obtain such results, it is fair to say that the results are novel and comprehensive (spanning many different question, mostly solved to completion).

**Weaknesses:**

The paper is densely written and assumes some prior knowledge of the literature from the get-go (for instance, the jump from concept classes from balls in graphs in the introduction is a bit too sudden without explaining how a set of balls in G form a binary concept class, especially when graph quantities starts appearing in the bounds (e.g. line 67)).

While this is expected from a theoretical paper, especially one that studies a rather niche problem, for the purpose of a machine learning conference, it is highly recommended that the authors defer more full proofs to the appendix, replacing it with a sketch proof, and use the extra space to motivate the relevance of the problem to the machine learning community (e.g. Line 108-114 can benefit from some illustration). A summary such as Table 1 is also something that should be in the main paper, at least the parts that are most significant.

Most of the proofs can use simple illustrations and better guidance (e.g. Theorem 2, Lemma 6, Lemma 10) because of how dense the notations and case analysis is.

In summary, the theoretical results are interesting and comprehensive, but the presentation of the paper needs major overhaul to be readable. I am willing to give another read and revise the score if the authors can clean up their expositions in the rebuttal period.

**Questions:**

n/a

---

> ### Author Response · Authors · 2025-11-21
>
> Thank you very much for your kind review and your helpful suggestions! We hope that the updated version of our paper has addressed your concerns.
>
> Concerning the comment about assuming some prior knowledge of the literature from the get-go: we are unsure what is meant by this since the first 3 pages of our submission extensively cover the state of the art from the literature in detail and compare our results to it. In fact, now we have also moved the table summarizing our results and those from the literature into these first 3 pages, so perhaps this has helped to address this comment. A sudden jump is also mentioned at line 67, however please note that lines 64-67 discuss open neighborhoods in graphs, which are not balls in graphs since the open neighborhood of a vertex $v$ (denoted by $N(v)$) does not contain $v$. In the next paragraph, we define balls in graphs and mention their equivalence with binary concept classes, but defer the explanation of the equivalence to the top of page 3 as it also holds for closed neighborhoods in graphs. However, the equivalence is not necessary for understanding any of our results or those from the literature.
>
> As per your suggestion, we have included a figure for the equivalence between balls in graphs and binary concept classes, and Table 1 now appears in the main text as well. We have also deferred part of the proof of Lemma 6 to the appendix to make room for these figures and other new ones.
>
> We feel that there is already a lot of content in the first 3 pages motivating the studied problem with respect to other areas of machine learning, machine teaching, and the other papers on non-clashing teaching. Thus, we have used the extra space to help the reader with the proofs by adding figures mainly. In particular, there are now figures for the case analyses in the proofs of Lemmas 6 and 10, there is a figure illustrating the pruning of the treedepth decomposition for Theorem 5, the figure illustrating a treedepth decomposition is now in the main text, and the figure for Theorem 14 is also now in the main text. Finally, as suggested by another reviewer, we have also added guided steps in both of the reduction figures (for Theorems 2 and 3) in the appendix to help the reader better understand the proofs just from the figures themselves.
>
> Further, we have simplified the statement (and thus proof) of Lemma 1 so that now it only covers our exact use of it. We have also made some minor adjustments to some of the other proofs in cases where we thought the reader could be aided a bit more.

---

> > ### Comment · Reviewer_BjtC · 2025-11-26
> >
> > I thank the authors for the rebuttal and the updated version. The extra illustrations are rather helpful but the exposition remains very dense for a general ML audience. Therefore I raised my score slightly as I think the content is still interesting.

---

> > > ### Author Response · Authors · 2025-11-26
> > >
> > > Thank you very much! We are glad that you appreciated the extra illustrations and find our work interesting.

---

### Author Response · Authors · 2025-11-21
**Changes in the updated version of the paper**

We thank all of the reviewers for their helpful suggestions!

In the updated version of our paper, we have improved the exposition by the following: we have added 1) a figure for the equivalence between balls in graphs and binary concept classes, 2) figures for the case analyses in the proofs of Lemmas 6 and 10, 3) a figure illustrating the pruning of the treedepth decomposition for Theorem 5, and 4) now Table 1, the figure illustrating a treedepth decomposition, and the figure for Theorem 14 are included in the main text. We have also added guided steps in both of the reduction figures (for Theorems 2 and 3) in the appendix to help the reader better understand the proofs just from the figures themselves. Further, we have simplified the statement (and thus proof) of Lemma 1 so that now it only covers our exact use of it. We have also made some minor adjustments to some of the other proofs in cases where we thought the reader could be aided a bit more. As suggested, we also put part of the reverse direction of the proof of Lemma 6 into the appendix, in order to make room for all of these figures. Thus, the extra page (and even a little more) that we are allowed for the rebuttal phase has completely gone to figures
and the table to help with the exposition as suggested by the reviewers.

---

> ### Author Response · Authors · 2025-11-26
>
> Dear reviewers,
>
> we thank you again for your hard work in delivering your reviews. We are fully aware that this was a lot of work to be done in a very short period. We have put a lot of effort into addressing your helpful suggestions and we would really appreciate it if you could have a look at our rebuttal and the updated version of the paper, and provide your feedback.

---

> ### Author Response · Authors · 2025-11-30
> **New version uploaded with color highlighting modifications (as requested)**
>
> We have just uploaded a new version where we have highlighted (in blue) the main modifications made during the rebuttal, as requested by one of the reviewers. For modified or added figures, we have simply highlighted the caption in blue. Our previous comment also details the modifications in order to help distinguish between added material (figures), modified material (figures and text), and material moved to and from the appendix (figures and text).

---

### Author Response · Authors · 2025-11-30
**Summary of rebuttal phase for new AC**

All of the reviewers have agreed that our paper has strong and interesting theoretical results. Some of them also mention that our results are novel, comprehensive, significant, highly relevant, and extend the state of the art, and that our proofs are interesting and well-written.

Their feedback revolves around two main points:

1. The presentation can be improved, especially by including illustrations to guide the reader.
2. The paper is purely theoretical.

We have addressed point 1. by adding many new figures, modifying existing figures, simplifying a lemma, and adding more details in some proofs, as mentioned in our previous comments (and highlighted in blue in the current version). Two of the reviewers responded to our rebuttal, and they both agreed that the new presentation style was effective, and thus, raised their scores.

We have addressed point 2. in our responses to the specific reviewers that brought up this point. In summary, as with many other past ICLR papers, our focus in this paper is purely theoretical, and given the number and complexity of the results, we feel that practical experiments are out of the scope. Furthermore, we feel that a complete theoretical analysis should be done before practical experiments for the empirical analysis to be most effective. This explains the focus of our paper, which moves us even closer to fully understanding where the boundary of algorithmic tractability lies for these problems. Also, concerning the comment that our paper may primarily appeal to the learning theory subcommunity, we feel that this is the case for most papers and their respective subcommunities at these ML conferences. We would like to highlight that learning theory is specifically mentioned as a relevant topic in the call for papers, and we believe that other theoreticians would also be interested in our paper.

---

### Meta-Review · Area_Chair_bz5T · 2026-01-18

**Summary:**

Reviewers broadly agree that the theoretical results are strong, interesting, and largely correct, and that the work advances the current understanding of tractability boundaries for these teaching-dimension problems.

The main concerns in the initial reviews were readability for a general ML audience. The authors substantially improved the presentation in the rebuttal. The paper remains theory-heavy, but given ICLR’s scope and the strength of the results, I recommend acceptance.

**Reviewer Concerns:**

The major concerns about clarity and presentation style were largely addressed. Regarding the concerns about "too much theory", given the scope of ICLR, it is not a big issue.

**Reviewer Scores:**

Two of the previous negative reviewers agreed to raise their scores to postive.

---

### Decision · Program_Chairs · 2026-01-26

Accept (Poster)